# Stable inheritance of H3.3-containing nucleosomes during mitotic cell divisions

Xiaowei Xu[1,2,3,4,5], Shoufu Duan[1,2,3,4,5], Xu Hua[1,2,3,4], Zhiming Li [1,2,3,4], Richard He[1,2,3,4] & Zhiguo Zhang [1,2,3,4✉]

Newly synthesized H3.1 and H3.3 histones are assembled into nucleosomes by different histone chaperones in replication-coupled and replication-independent pathways, respectively. However, it is not clear how parental H3.3 molecules are transferred following DNA replication, especially when compared to H3.1. Here, by monitoring parental H3.1- and H3.3-SNAP signals, we show that parental H3.3, like H3.1, are stably transferred into daughter cells. Moreover, Mcm2-Pola1 and Pole3-Pole4, two pathways involved in parental histone transfer based upon the analysis of modifications on parental histones, participate in the transfer of both H3.1 and H3.3 following DNA replication. Lastly, we found that Mcm2, Pole3 and Pole4 mutants defective in parental histone transfer show defects in chromosome segregation. These results indicate that in contrast to deposition of newly synthesized H3.1 and H3.3, transfer of parental H3.1 and H3.3 is mediated by these shared mechanisms, which contributes to epigenetic memory of gene expression and maintenance of genome stability.

[1] Institute for Cancer Genetics, Columbia University Irving Medical Center, New York, NY, USA. [2] Herbert Irving Comprehensive Cancer Center, Columbia University Irving Medical Center, New York, NY, USA. [3] Department of Pediatrics, Columbia University Irving Medical Center, New York, NY, USA. [4] Department of Genetics and Development, Columbia University Irving Medical Center, New York, NY, USA. [5] These authors contributed equally: Xiaowei Xu, Shoufu Duan. ✉email: zz2401@cumc.columbia.edu

The nucleosome particle is the basic unit of eukaryotic chromatin, which encodes epigenetic information and maintains both genome and epigenome integrity. The nucleosome is composed of ~147 base pairs of DNA wrapped around a histone octamer containing two copies of each of the core histone proteins H2A, H2B, H3, and H4[1,2]. These histones are modified post-translationally, with different modifications marking different chromatin domains. For instance, tri-methylation of histone H3 lysine 4 is enriched at actively transcribed genes[3], whereas tri-methylation of histone H3 lysine 9 marks the silent heterochromatin[4,5]. In addition to histone modifications, chromatin is also demarcated by histone variants, which adopt similar structural folds to canonical histones and localize at specific regions of chromatin[6,7]. For instance, in dividing cells, while the majority of nucleosomes contain the canonical histones H3.1/H3.2, about 10–20% of nucleosomes utilize the H3 variant, H3.3[8]. H3.3 differs from H3.1/H3.2 by four or five amino acids and is enriched at actively transcribed genes, but is also localized at heterochromatin regions[9–11]. During S phase of the cell cycle, distinct chromatin states, marked by different histone modifications and variants, must be propagated into daughter cells to maintain gene expression state and cell identity, a process that remains elusive[12–14].

Following DNA replication, replicated DNA is assembled into nucleosomes using both newly synthesized histones and parental histones. It is known that newly synthesized and parental histone H3–H4 tetramers form distinct nucleosomes following DNA replication[8], suggesting that once assembled into nucleosomes, parental H3–H4 including H3.1 and H3.3 do not exchange freely with newly synthesized H3–H4. Newly synthesized H3.1, along with H4, are incorporated during the S phase in a process coupled with DNA replication. In this process, it is known that H3.1–H4 dimers first bind to Asf1 (Anti-Silencing Factor 1), and are then transferred from Asf1 to the downstream histone chaperone CAF-1 (Chromatin Assembly Factor 1) complex for deposition onto replicating DNA and subsequent nucleosome formation[9,15–18]. In contrast to H3.1, newly synthesized histone H3.3–H4 proteins are assembled into nucleosomes throughout the cell cycle[19]. It has been observed that new H3.3 can be incorporated into chromatin in the S phase, likely filling the gap left by H3.1–H4[20]. Moreover, two different histone chaperones facilitate nucleosome assembly of new H3.3–H4 tetramers. H3.3 histone chaperone Hira (Histone Regulator A) deposits H3.3–H4 at genic regions, including gene bodies, likely through its interaction with RNA polymerase II, and at promoters and enhancers through Hira-RPA (Replication Protein A) interaction[21,22]. In addition to Hira, Daxx (death domain-associated protein 6) and Atrx (alpha thalassemia X-linked intellectual disability) are also involved in deposition of new H3.3, likely at telomeric heterochromatin and endogenous retroviral elements[10,23–26].

In contrast to de novo deposition of new H3–H4, the transfer of parental histones behind DNA replication forks is relatively less well understood. Recent studies in budding yeast indicate that parental H3–H4, when transferred onto replicating DNA, can remember their positions following DNA replication and gene transcription[27]. In mouse embryonic stem (ES) cells, H3.1–H4 tetramers are also transferred locally at heterochromatin regions but are more dispersed at actively transcribed regions where H3.3-containing nucleosomes are enriched[28]. However, it is not known whether H3.3-containing nucleosomes at the actively transcribed regions can also remember their positions in mammalian cells, like in yeast cells.

Two pathways have been uncovered for the transfer of parental histones H3–H4 onto replicating DNA strands, and these pathways are conserved from yeast to human cells[29–32]. For instance, by monitoring histone modifications on parental histones as well as newly synthesized histones using the eSPAN (Enrichment and Sequencing of Protein-Associated Nascent DNA), it has been shown that yeast Dpb3 and Dpb4 (Pole4 and Pole3 in mammalian cells, respectively) are involved in the transfer of parental histones to leading strands, whereas the Mcm2-Ctf4-Polα axis functions in the transfer of parental histones to lagging stands at DNA replication forks[29,32]. The role of Mcm2 in parental histone transfer in mouse ES cells has also been independently reported using SCAR-seq (Sister Chromatids After Replication by DNA sequencing)[30]. Moreover, Dpb3/Pole4 and Dpb4/Pole3, which form a dimer with a structure similar to that of histone H2A-H2B[33], bind H3–H4[32,34]. Additionally, both Mcm2 and Pol1 (Pola1 in mammalian cells) contain a histone binding motif that interacts with H3–H4[31,35]. These studies reveal that several replisome components function as histone chaperones for the recycling of parental histones following DNA replication. However, most of these studies in mammalian cells monitored modification on histones including H4K20me2 and H3K36me3, which are likely present on both H3.1- and H3.3-containing nucleosomes. Therefore, it remains to be determined whether parental H3.1 and H3.3 utilize different factors, like their newly synthesized counterparts, for their transfer to replicating DNA strands during DNA replication.

In recent studies using SNAP-tagged H3.1 and H3.3 to monitor parental H3.1 and H3.3 recycling in HeLa cells, it has been observed that both H3.1 and H3.3 are lost at a higher rate than what have expected from S phase dilution[36,37], with H3.3 losing more rapidly than H3.1. However, these studies did not monitor changes in histone levels within each cell during one cell cycle. In this report, we utilized live-cell imaging and monitored SNAP-tagged parental histone H3.1 and H3.3 throughout one cell cycle. We observe that surprisingly, both parental H3.1 and H3.3 are stably recycled during one cell division and segregate into two daughter cells in almost equal amounts. Moreover, we show that the Mcm2-Pola1 and Pole3–Pole4 pathways are involved in the transfer of both H3.1 and H3.3 following DNA replication. Finally, we observed defects in chromosome segregation in Mcm2, Pole3, and Pole4 mutant cells with impaired parental histone transfer. Together, our findings reveal that the DNA replisome components Mcm2, Pola1, Pole3, and Pole4 also function as histone chaperones for parental H3.3 during the S phase of the cell cycle, safeguarding both genome and epigenome integrity during mitotic cell division.

## Results

**Both parental histone H3.1 and H3.3 are stably recycled during mitotic cell division**. It is known that newly synthesized H3.1 and H3.3 are assembled into nucleosomes via distinct histone chaperones in a DNA replication-dependent and replication-independent process, respectively[9]. However, it remains unclear whether H3.1- and H3.3-containing parental nucleosomes are recycled via similar or distinct mechanisms during mitotic cell division. To monitor the segregation of parental H3.1 and H3.3 into daughter cells, we first employed CRISPR/Cas9 genomic editing technology to fuse the SNAP tag at the C-terminus of H3.1 (*Hist1h3g*) and H3.3 (*H3f3b*) (Supplementary Fig. 1a, b). The SNAP tag is a modified form of O-6-methylguanine-DNA methyltransferase that can covalently attach O6-benzylguanine derivatives. We verified the expression of H3.1-SNAP and H3.3-SNAP fusion proteins using western blot (Supplementary Fig. 1c). Moreover, we found that expression of H3.1-SNAP or H3.3-SNAP did not alter the karyotype or cell cycle of each tagged cell lines (Supplementary Fig. 1d–g).

Next, we labeled H3.1-SNAP or H3.3-SNAP with the fluorescent substrate TMR for 30 min. After washing out TMR

substrates, we monitored the distribution of H3.1-SNAP and H3.3-SNAP in individual cells for 16 h at 20 min intervals using live-cell fluorescence microscopy. In this way, we can monitor the distribution of parental H3.1-SNAP and H3.3-SNAP in individual cells during the cell cycle progression. To analyze the distribution of parental H3.1-SNAP and H3.3-SNAP during the progression of the cell cycle, we first identified individual cells at mitotic cell stage based on the compaction of chromosomes and used this as the reference point to define G1/S (11 h before mitosis), G2 (1 h before mitosis), and next G1 (1 h after mitosis). Please note that this estimation was confirmed using cell cycle indicator in wild type and each of the mutant cells described below (see Fig. 4). We then quantified the integrated TMR signals in each individual cell at three time points, G1/S, G2, and subsequent G1 (Fig. 1a). Based on the integrated TMR signals of each individual cell, as well as the average of the integrated TMR signals of all cells analyzed, we found that the amount of parental H3.1 at G2 was the same as at G1/S, indicating that canonical parental H3.1 are faithfully recycled following DNA replication (Fig. 1b, c, Supplementary Movie 1). Moreover, the two daughter cells inherited an equal amount of parental H3.1 from the mother cell, with each receiving about half of the H3.1-SNAP proteins compared to G1/S or G2 cells. Furthermore, the sum of the integrated TMR signals of two daughter cells was the same as that at G2 (Fig. 1b, c). Taken together, these results are consistent with the idea that the vast majority of parental H3.1 proteins are recycled following DNA replication, with each of the daughter cells receiving an equal amount of parental H3.1.

Surprisingly, similar analysis on the distribution of H3.3-SNAP proteins in cells at G2 and G1/S transition following DNA replication revealed that parental H3.3 were also largely stably maintained following DNA replication and equally distributed into two daughter cells (Fig. 1d, e, Supplementary Movie 2). It is possible that H3.3-SNAP mRNA and proteins are more stable than wild-type H3.3, which in turn contributes to the stable inheritance of H3.3 following DNA replication. To test this possibility, we first compared the expression level of H3f3b-SNAP with H3f3b in wild-type ES cells using RT-PCR and found that the expression of the SNAP-tagged H3f3b was similar to wild-type H3f3b (Supplementary Fig. 2a). Next we measured the decay rate of nascent RNA pulse labeled with 4sU and observed that the half-life of WT H3f3b and H3f3b-SNAP mRNA was quite similar (Supplementary Fig. 2b). Taken together, these results indicate that the SNAP tag did not affect the stability of H3f3b mRNA. Finally, we estimated that the protein levels of H3.3-SNAP and H3.3 expressed from *H3f3a*, another gene encoding H3.3, by Western blot and found that the levels of H3.3-SNAP were lower than H3.3 (Supplementary Fig. 2c, d). We noticed that in mouse ES cells, the expression of *H3f3b* is about 1.7-fold of *H3f3a* based on RNA-seq analysis, suggesting that H3.3-SNAP tag did not increase the stability of H3.3 proteins. Taken together, our results collectively show that both parental H3.1 and H3.3 proteins are largely recycled during DNA replication, with each daughter cell receiving about half of both H3.1 and H3.3 from the mother cell following mitotic cell division.

We also noticed that while there was no significant difference between the average integrated H3.1- and H3.3-SNAP signals at G2 and combined signals of two G1 daughter cells, there was a small, but statistically significant reduction of combined H3.1-SNAP signals of two G1 daughter cells compared to their corresponding mother cells at G1/S. This reduction, likely due to bleaching of fluorescent signals overtime, was detected for four independent repeats of H3.1-SNAP signals in wild-type cells (see Figs. 1 and 2), and 6 out 8 independent repeats of H3.3-SNAP (Figs. 3, 4 and Supplementary Fig. 4, but not in Fig. 1). Therefore, in the rest of our studies, we will compare H3.1-SNAP and

H3.3-SNAP signals at G2 with those of G1/S transition, which did not show statistical differences in wild-type mouse ES cells in all our experiments to discern the impact of gene mutations on parental histone segregation following DNA replication.

**Mutations in genes involved in parental histone transfer impair parental H3.1 recycling.** By monitoring histone modifications on parental H3 at replicating DNA strands, we and others have shown that mutations in Mcm2, a subunit of the replicative helicase MCM, as well as in Pole3 and Pole4, two subunits of the leading strand DNA polymerase Polε, affect parental histone transfer to lagging and leading strands of DNA replication forks, respectively[29–32]. However, these studies did not test to what extent the overall levels of parental H3.1 and H3.3 proteins are affected in these mutant cells after one cell division. As most parental histone H3 in mouse ES cells are the canonical histone variants H3.1/H3.2, we first analyzed the effects of Pole4 deletion and Mcm2-2A mutation on overall levels of histone H3.1 using live-cell immunofluorescence. Pole4 KO, Mcm2-2A, and Mcm2-2A Pole4 KO double mutations were introduced in H3.1-SNAP cells and confirmed by Sanger sequencing (Supplementary Fig. 3a). These mutations had no apparent effects on overall level of H3.1-SNAP (Supplementary Fig. 3b). We observed that the average H3.1-SNAP (TMR) signals at the G2 phase in Pole4 KO, Mcm2-2A, and Mcm2-2A Pole4 KO double mutant cells were significantly reduced compared to that of G1/S, in contrast to the unreduced levels observed in wild-type cells (Fig. 2a, b, Supplementary Movies 3–6). Moreover, H3.1-SNAP signals at G2 were similar to the sum of two daughter cells at G1 (Fig. 2b). The impact of these mutants on H3.1-SNAP signals at the G2 phase compared to G1/S was also apparent from analysis of the integrated H3.1-SNAP signal in individual cells (Supplementary Fig. 3c). In contrast, in wild-type cells, H3.1-SNAP signals at G2 phase were similar to those at G1/S phase in the same experimental settings to analyze the effects of each mutant using live-cell image analysis (Fig. 2a, b). These results indicate a slight but statistically significant loss of parental histone H3.1 in Pole4 KO, Mcm2-2A single, and double mutant cells during the passage of S phase. Consistent with this idea, we also observed that the integrated H3.1-SNAP signals at G2 in the mutant cells were significantly lower than those in wild-type cells (Fig. 2c). We noticed that the effect of Mcm2-2A Pole4 KO double mutation on overall levels of parental H3.1-SNAP was similar to Mcm2-2A or Pole4 KO single mutant alone, likely due to insensitivity of the live-cell imaging assays that may not able to detect differences in histone recycling between Mcm2-2A Pole4 KO double mutant cells and Mcm2-2A or Pole4 KO single mutant cells. Alternatively, we noticed that Mcm2-2A Pole4 KO double mutant cells showed increased apoptosis compared to Mcm2-2A and Pole4 KO single mutant alone (Supplementary Fig. 3d). The increase in apoptosis in the double mutant cells likely reflects dramatic defects in the recycling of parental H3–H4 in the double mutant cells, thereby contributing to the inability to detect the differences in recycling of parental H3.1-SNAP between Mcm2-2A Pole4 KO double mutant cells and Mcm2-2A or Pole4 KO single mutant. Collectively, our results indicate that Pole4 and Mcm2 participate in the faithful recycling of parental histone H3.1 during S phase of the cell cycle.

**Mcm2 and Pole3/4 are also required for recycling of parental histone H3.3.** Hira and Daxx are two histone chaperones involved in nucleosome assembly of newly synthesized H3.3 through all phases of the cell cycle, and CAF-1 deposits H3.1 onto replicating DNA during S phase of the cell cycle[16,21,23,25]. Moreover, Hira has been implicated as an important player in

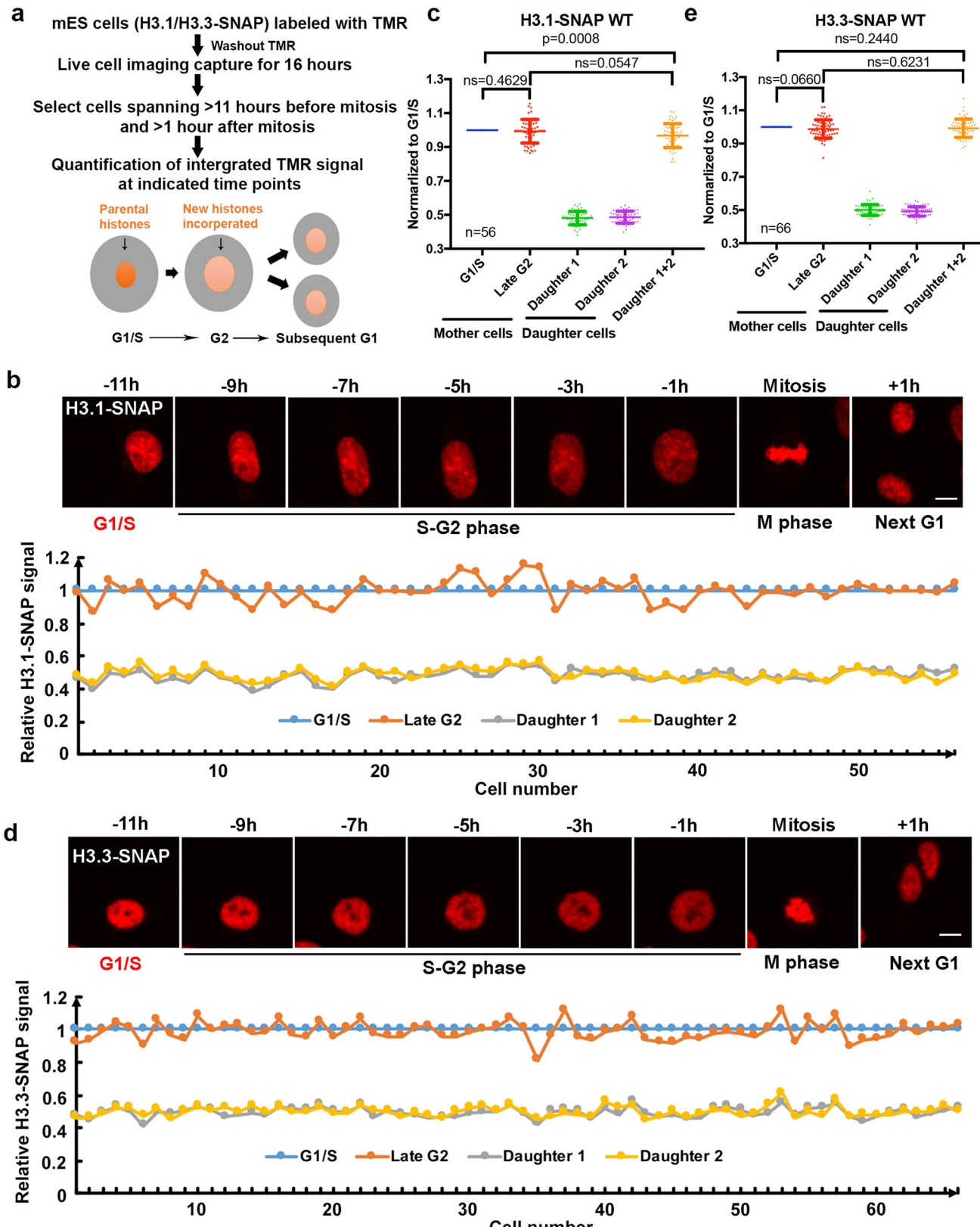

recycling parental H3.3 during gene transcription[36]. However, it is not known whether Hira and Daxx are also required for recycling parental H3.3 during the S phase of the cell cycle. Therefore, we deleted Hira and Daxx in the H3.3-SNAP cell line (Supplementary Fig. 4a, b) and monitored parental H3.3 dynamics during cell cycle progression. We observed that like wild-type cells, H3.3-SNAP signals at the G2 phase were the same as those at G1/S in Hira KO cells, suggesting that Hira deletion has no detectable impact on recycling parental H3.3 during the S phase of the cell cycle. Conversely, the average integrated H3.3-SNAP signals at G2 were reduced compared to those at G1/S in Daxx KO cells (Supplementary Fig. 4c–g, Supplementary

**Fig. 1 Parental H3.1 and H3.3 are faithfully recycled following one cell division. a** An outline of experimental procedures for the quantification of parental H3.1 and H3.3 during one cell division. H3.1-SNAP or H3.3-SNAP-tagged cells were labeled with TMR. After washing out TMR substrates, live-cell images were captured continuously for 16 h. Cells that showed TMR signals more than 11 h before mitosis (G1/S) and more than 1 h after mitosis (next G1) were chosen for analysis of the integrated intensity of H3.1-SNAP and H3.3-SNAP signals at G1, G2, and next G1. **b** Parental H3.1 are stably recycled following DNA replication and equally distributed to two daughter cells. Upper: representative live-cell images of TMR-labeled-parental H3.1 at the indicated time points in mES cells expressing H3.1-SNAP (*n* = 56). The number denotes time in hours in reference to mitosis. Scale bar, 10 μm. Lower: integrated TMR signals normalized to G1/S time point at three time points: G1/S, Late G2 and next G1 with two daughter cells. Relative TMR signals at G1/S, G2 in each individual cell and those in daughter 1 and 2 are shown. **c** Quantification of parental H3.1-SNAP signals at different phases of the cell cycle. Parental H3.1-SNAP fluorescence in each cell was measured in the entire nucleus at G1/S, G2, daughter cell 1 and 2 and normalized to that mother cell at G1/S. **d** Parental H3.3 are stably recycled following DNA replication and equally distributed to two daughter cells. Upper: representative live-cell images of parental H3.3 at the indicated time points in cells expressing H3.3-SNAP (*n* = 66). Scale bar, 10 μm. Lower: relative TMR signals at G1/S, G2 in each individual cell and those in daughter 1 and 2 are shown. **e** Quantification of parental H3.3 signals at different phases of the cell cycle. The experimental procedures mirror that described in (**c**). **b**–**e** *n* number of cells from two independent experiments. **c**, **e** Data are presented as means ± SD. Two-tailed unpaired Student *t* test were performed with the *P* values marked on the graphs (ns, no significant difference).

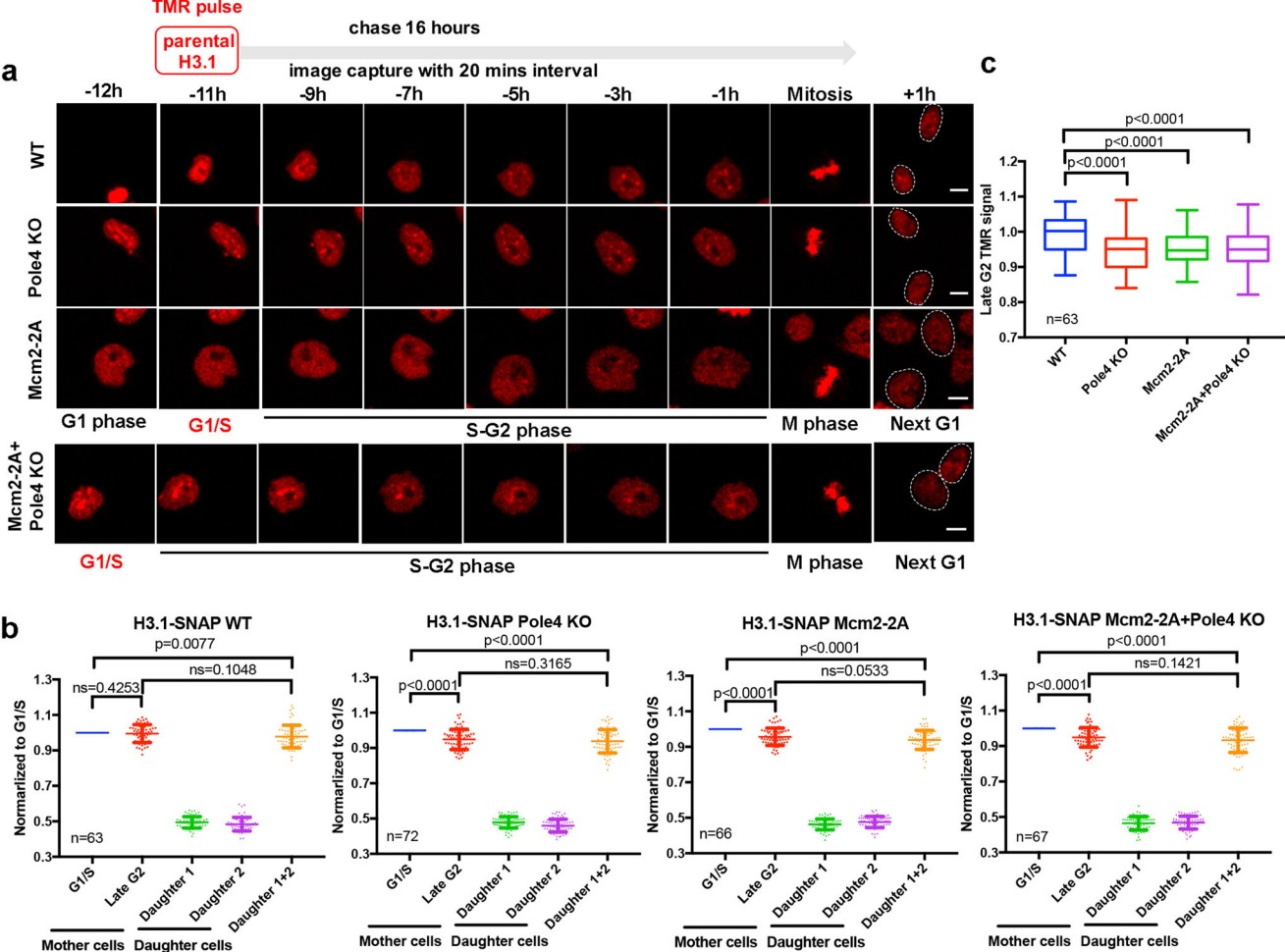

**Fig. 2 Histone chaperones Mcm2 and Pole4 are involved in the recycling of parental H3.1. a** Upper: an experimental scheme for the analysis of H3.1-SNAP in mES cells using live-cell imaging. Lower: representative live-cell images of TMR of parental histone H3.1-SNAP at the indicated time points in WT (*n* = 63), Pole4 KO (*n* = 72), Mcm2-2A (*n* = 66) and Pole4 KO + Mcm2-2A double mutant (*n* = 67) mES cells. Scale bar, 10 μm. Two daughter cells arising from the mother cell were circled. **b** Quantification of parental H3.1-SNAP signals WT, Pole4 KO, Mcm2-2A and Mcm2-2A + Pole4 KO mES cells at G1/S and G2 of each mother cell and G1 of the two individual daughter cells. Data are presented as means ± SD. **c** Boxplot of relative parental H3.1-SNAP signals at G2 calculated in (**b**) among WT, Pole4 KO, Mcm2-2A and Mcm2-2A + Pole4 KO mES cell lines. The center line is the medians of all data points, with the limits corresponding to the upper and the lower quartiles, respectively, and the whiskers representing the largest and smallest values.
**a**–**c** *n* number of cells from two independent experiments. **b**, **c** Statistical analysis was performed by two-tailed unpaired Student *t* test with *P* values shown on the graphs. ns no significant difference.

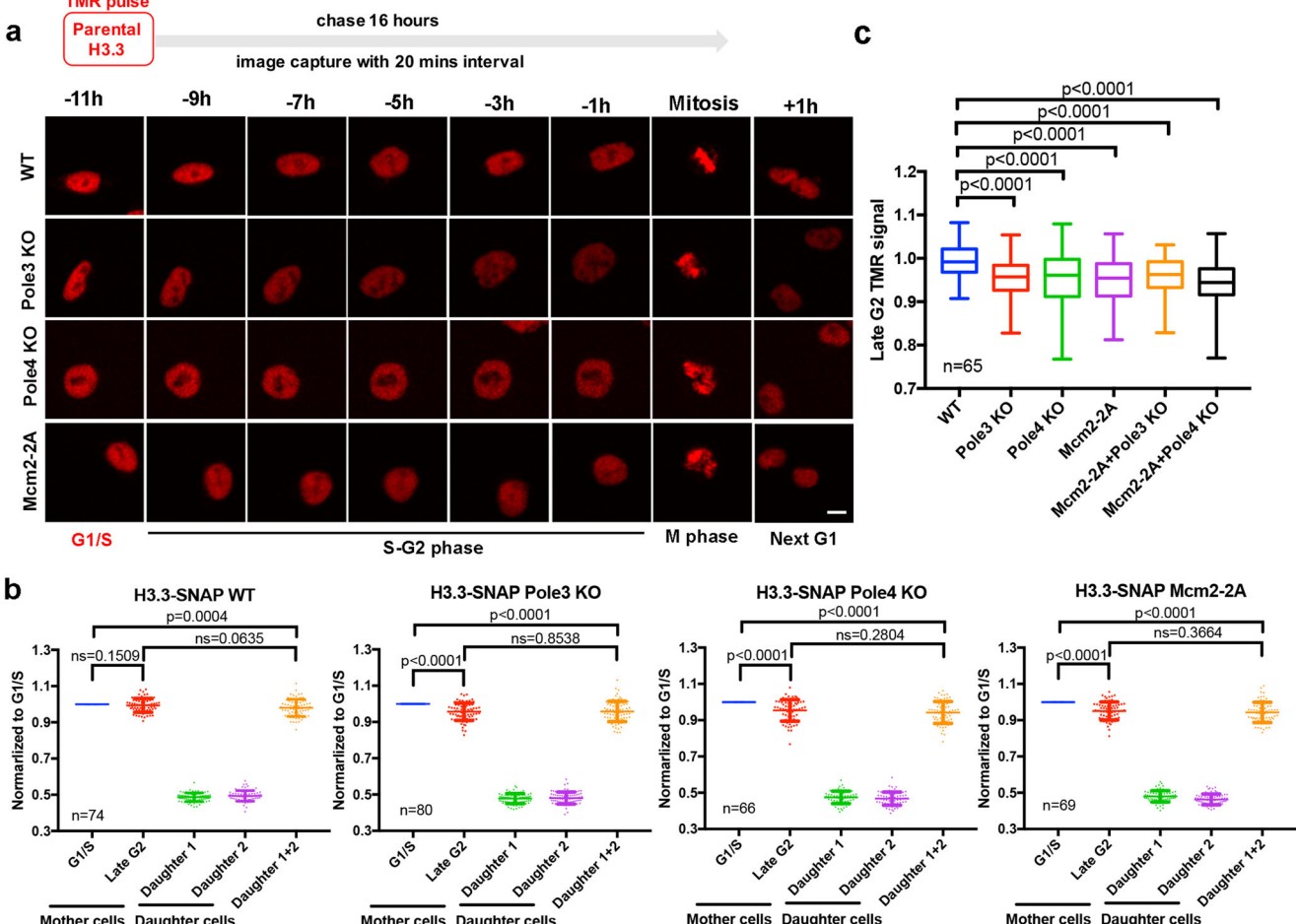

**Fig. 3 Mcm2, Pole3, and Pole4 are required for faithfully parental histone H3.3 recycling during S phase. a** Representative live-cell images of parental histones H3.3-SNAP signals at the indicated time points in WT (*n* = 74), Pole3 KO (*n* = 80), Pole4 KO (*n* = 66), and Mcm2-2A (*n* = 69) mES single mutant cell lines. Scale bar, 10 μm. **b** Quantification of H3.3-SNAP signals at each individual cell of WT, Pole3 KO, Pole4 KO, and Mcm2-2A single mutant mES cells. Data are presented as means ± SD. **c** Comparison of the relative amount of parental H3.3-SNAP at individual cells at late G2 in WT, Pole3 KO, Pole4 KO, Mcm2-2A, Mcm2-2A + Pole3 KO, and Mcm2-2A + Pole4 KO mES cell lines. The center line, the box limits and the whiskers are defined as described in Fig. 2c. **a–c** *n* number of cells from two independent experiments. **b, c** Two-tailed unpaired Student *t* tests were performed with the *P* values marked on the graphs, and ns no significant difference.

Movies 7–9). These results indicate that while Hira likely has limited roles in recycling parental H3.3 during S phase of the cell cycle, Daxx may function in both deposition of new H3.3 and recycling of parental H3.3.

By monitoring H3K36me3 at replicating DNA strands using eSPAN, we have shown that Mcm2, Pola1, Pole3 and Pole4 are involved in the transfer of this modified form of H3 onto replicating DNA strands[31]. Because H3K36me3 is enriched at H3.3 compared to H3.1[38], it is possible that Mcm2 and Pole3/Pole4 also function in recycling parental H3.3. To test this hypothesis, we first generated Mcm2-2A, Pole3 KO and Pole4 KO cell lines in H3.3-SNAP cells (Supplementary Fig. 5a, b). We then analyzed the distribution of H3.3-SNAP during cell cycle progression via live-cell fluorescence microscopy followed by the quantification of H3.3-SNAP signals at G1/S, G2 and next G1 of two daughter cells as described in Fig. 1. By analyzing the average integrated signals of individual cells (Fig. 3a, b, Supplementary Movies 10–13) as well as at individual cells of each indicated line (Supplementary Fig. 5d), we observed a slight but statistically significant decrease in parental histone H3.3 signals at the G2 phase in Mcm2-2A, Pole3 KO and Pole4 KO cells relative to those at the corresponding G1/S phase, whereas in wild-type cells H3.3-SNAP signals at G2 phase are

similar to those at G1/S phase. Furthermore, we also observed that the relative amount of H3.3-SNAP at G2 phase was reduced in each mutant compared to the wild type (Fig. 3c). Finally, double mutants containing Mcm2-2A and either Pole3 KO or Pole4 KO mutations also showed defects in parental histone recycling (Supplementary Fig. 5c–f, Supplementary Movies 14, 15) to a similar degree as each of the single mutants (Supplementary Fig. 5c–f, Fig. 3c). Taken together, these results indicate that Mcm2, Pole3 and Pole4 are involved in recycling parental histone H3.3 during the S phase of the cell cycle.

**Pola1 also facilitates parental histone H3.3 recycling.** We have shown previously that mutations at histone binding motif of Pola1, the catalytic subunit of mammalian Polα primase, also affects parental histone transfer to lagging strands of DNA replication forks[31]. To provide additional evidence supporting the idea that factors involved in parental histone recycling are also important for recycling of H3.3 following DNA replication, we decided to analyze the effect of Pola1-2A mutant on H3.3-SNAP signals during cell cycle progression using live-cell image. Furthermore, the G1/S transition in the experiments presented above was estimated based on cell cycle progression. To more precisely identify G1/S transition and to eliminate the potential cell cycle

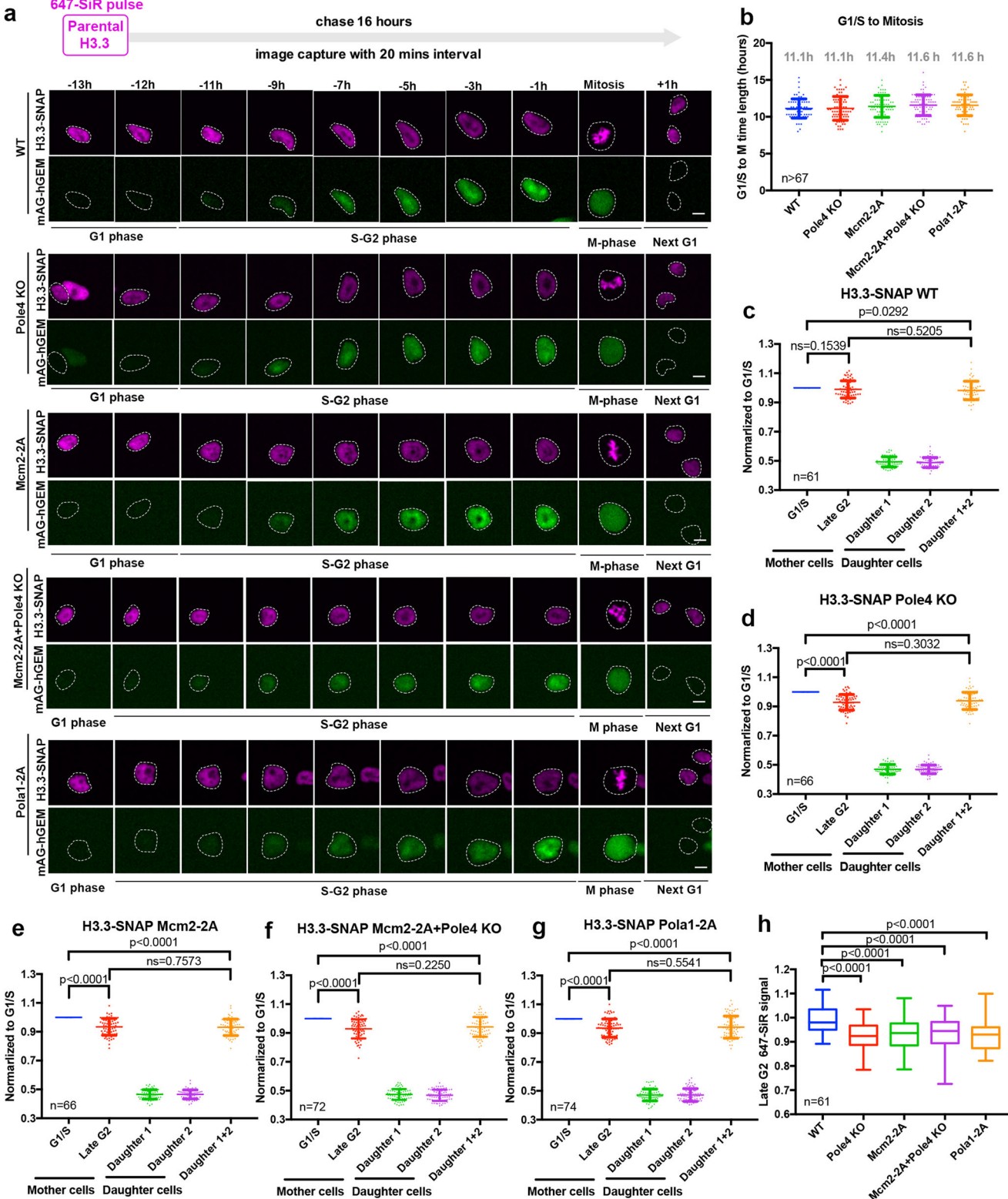

effects between WT and each of the mutant cells on H3.3-SNAP recycling, we introduced the fluorescence ubiquitination-based cell cycle indicator (FUCCI)[39] into wild type, Mcm2-2A, Pole4 KO, Pola-2A single, and Mcm2-2A Pole4 KO double mutant cells. This system relies on the degrons of two proteins, Cdt1 and its inhibitor, geminin, which are fused to mKO2 and mAG, respectively. Cdt1 and Geminin are involved in DNA replication control. Moreover, their expression is regulated by ubiquitin-mediated degradation, with the expression of Cdt1 peaking during G1 phase, and the expression of geminin peaking in S and G2 phase, but being low in late mitosis and G1 phase[39]. Because the expression of mKO2-Cdt1 was low, we decided to use live-cell image to monitor both the expression of mAG-Geminin and parental H3.3-SNAP signals during cell cycle progression to avoid additional complications from photobleaching of fluorescence signals over time. First, we analyzed the average time

**Fig. 4 Pola1 facilitates parental histone H3.3 recycling. a** Representative live-cell images of parental histones H3.3-SNAP signals at the indicated time points in WT ($n = 61$), Pole4 KO ($n = 66$), Mcm2-2A ($n = 66$), Pola1-2A ($n = 74$) mES single and Mcm2-2A Pole4 KO double mutant ($n = 72$) cell lines. Purple channels are 647-SiR labeled-parental H3.3-SNAP, green channels are for mAG tagged-geminin expression. Scale bar, 10 μm. Tracked cells were circled. **b** The average time from G1/S to mitosis for WT, Pole4 KO, Mcm2-2A, Pola1-2A, and Mcm2-2A Pole4 KO double mutant mES cell lines based on the appearance of mAG tagged-geminin (G1/S) and the reduction of mAG tagged-geminin signals (mitosis). **c–g** Quantification of H3.3-SNAP signals at G1/S and G2 of each mother cell and G1 of its two daughter cells in WT, Pole4 KO, Mcm2-2A, Pola1-2A, and Mcm2-2A + Pole4 KO mutant mES cell lines. The cell cycle stage of each cell was based on mAG tagged-geminin signals. **h** Comparison of the relative amount of parental H3.3-SNAP at individual cells at late G2 in WT, Pole4 KO, Mcm2-2A, Pola1-2A and Mcm2-2A + Pole4 KO mES cell lines. The center line, the box limits and the whiskers were defined as described in Fig. 2c. **b–g** Data are presented as means ± SD. **a–h** $n$ number of cells from two independent experiments. **c–h** Statistical analysis was performed by two-tailed unpaired Student $t$ test with the $P$ values marked on the graphs, ns no significant difference.

needed for WT, Pole4 KO, Mcm2-2A, Pola1-2A single and Mcm2-2A Pole4 KO double mutant cells transition from G1/S to mitosis based on the appearance (G1/S) and the reduction of mAG-Geminin (mitosis) signals during cell cycle progression. We found that it took about 11 h for wild type, Pole4 KO, and Mcm2-2A cells to transition from G1/S to mitosis. However, it took about 11.6 h for Mcm2-2A Pole4 KO double mutant cells to transition from G1/S to mitosis (Fig. 4b). These results indicate that the time points (11 h before mitosis in single mutants and 12 h before mitosis in double mutants) used to estimate the G1/S transition in wild type and each mutant cells in Figs. 2 and 3 are sound.

Next, we quantified parental H3.3-SNAP integrated signals at G1/S based on the appearance of geminin expression in each cell and compared to those at G2 phase. We observed that in wild-type cells H3.3-SNAP signals at G2 phase were similar to those at G1/S phase (Fig. 4a, c). In contrast, parental H3.3-SNAP signals at the G2 phase in Mcm2-2A, Pole4 KO single and Mcm2-2A Pole4 KO double mutant cells were lower at G2 phase than the corresponding ones at G1/S based on the analysis of average integrated H3.3-SNAP signals (Fig. 4c–f) and as well as at individual cells of each indicated line (Supplementary Fig 6, Supplementary Movies 16–19). Similarly, we also observed parental H3.3 recycling defects in Pola1-2A mutant cells (Fig. 4g, Supplementary Movies 20). Furthermore, we observed that the relative amount of H3.3-SNAP at G2 phase was reduced in each mutant cells compared to wild type (Fig. 4h). Taken together, this independent analysis of H3.3-SNAP signals at different cell cycle phase defined by mAG-geminin provide additional evidence supporting the idea that Mcm2-Pola1 and Pole3–Pole4 participate in the recycling of parental histone H3.3 during S phase of the cell cycle.

**The Mcm2-Pola1 axis and Pole3–Pole4 facilitate parental H3.3 transfer to lagging and leading strands, respectively.** It has been shown that Mcm2 binds to H3.3 in vitro[40]. To understand how Mcm2, Pole3 and Pole4 function in recycling of H3.3, we first analyzed whether these three proteins bind to H3.3 in vivo. Briefly, we generated Flag-tagged H3.3 mES cells by CRISPR/Cas9 genome editing (Supplementary Fig. 7a, b) and immunoprecipitated H3.3-Flag from cell extracts. Both Mcm2 and Pole4 coimmunoprecipitated with H3.3-Flag, confirming that they interact with histone H3.3 in vivo (Fig. 5a). Next, we tested whether Mcm2-2A mutants defective in histone binding also display defects in binding to H3.3. To this end, we tagged Mcm2 WT and Mcm2-2A mutant proteins with the Flag tag in cell lines expressing H3.3-SNAP (Supplementary Fig. 7a, c) and immunoprecipitated Flag-Mcm2 WT and Flag-Mcm2-2A proteins. We observed that H3.3-SNAP co-purified with Mcm2 WT, consistent with the idea that Mcm2 interacts with H3.3. Importantly, the observed Mcm2-H3.3 interaction was markedly reduced in Mcm2-2A cells compared to Mcm2 WT (Fig. 5b), providing an explanation for the compromised recycling of parental H3.3 in Mcm2-2A cells (Figs. 3 and 4).

We have monitored two post-translational modifications (PTMs) that mark parental histones, H4K20me2 or H3K36me3, using the eSPAN method, which measures the relative amount of these modified forms of H3 at leading and lagging strands of DNA replication forks, in mES cells[31]. We discovered that Mcm2-Pola1 and Pole3/Pole4 are involved in the transfer of parental H3–H4 onto lagging and leading strands of DNA replication forks, respectively. These histone marks are likely present on both parental H3.1 and H3.3. To determine whether Mcm2-Pola1 and Pole3–Pole4 are involved in the transfer of parental H3.3 onto replicating DNA strands, we first labeled H3.3-SNAP with the SNAP-biotin substrates. After washing away the SNAP-biotin substrates, we cultured these cells for 14 h (Fig. 5c), thus ensuring that nearly all of the biotin labeled H3.3-SNAP proteins are parental histones. We then pulsed these cells for 40 min with BrdU, a nucleotide analog incorporated into newly synthesized DNA, and performed CUT&Tag with biotin specific antibodies. Tagmented DNA was denatured and immunoprecipitated with BrdU antibody (eSPAN), followed by library preparation and sequencing (Fig. 5c).

Consistent with published data, H3.3-SNAP CUT&Tag signals were enriched at the TSS and gene bodies of highly transcribed genes compared to those in lowly transcribed genes (Fig. 5d, e), implying that biotin-H3.3-SNAP CUT&Tag generates reliable H3.3 chromatin localization profiles. Moreover, this distribution was not affected in Mcm2-2A, Pola1-2A, and Pole3/4 KO cells (Supplementary Fig. 7d), and H3.3-SNAP CUT&Tag peaks did not show any bias towards leading or lagging strands (Supplementary Fig. 7e). Next, we analyzed the distribution of H3.3-SNAP eSPAN surrounding 1,548 origins we previously identified in mES cells[31]. We observed that H3.3-SNAP eSPAN signals in WT cells showed a slight bias towards leading strands (Fig. 5f–i). Notably, H3.3-eSPAN in Mcm2-2A and Pola1-2A mutants showed a marked increase in leading strand bias compared to WT (Fig. 5f, g, i), supporting the idea that parental H3.3 transfer to lagging strand was compromised in these two mutant cells. In contrast, we observed that the H3.3-SNAP eSPAN leading strand bias in Pole3 or Pole4 KO cells was reduced when compared to wild-type cells (Supplementary Fig. 7f). Finally, we observed that the bias of H3.3-SNAP and H3K36me3 eSPAN signals in Mcm2-2A Pole4 KO double mutant cells was similar to that of WT cells (Fig. 5h, i, Supplementary Fig. 7g), consistent with the idea that Mcm2 and Pole4 function to transfer parental H3–H4 including H3.3 to lagging and leading strands, respectively, rendering the eSPAN bias in Mcm2-2A Pole4 KO double mutant cells to that of wild-type cells. We noticed, however, that the effects of Mcm2-2A, Pola1-2A, and Pole4 KO on H3.3-SNAP eSPAN bias were markedly smaller than the impact of these mutations on H3K36me3 eSPAN bias (Supplementary Fig. 7g, h). One possible explanation for this differential effect on the distribution of H3.3-SNAP and H3K36me3 at replicating DNA strands is that the 1,548 replication origins are localized at actively transcribed regions where parental H3.3 transferred to replicating DNA

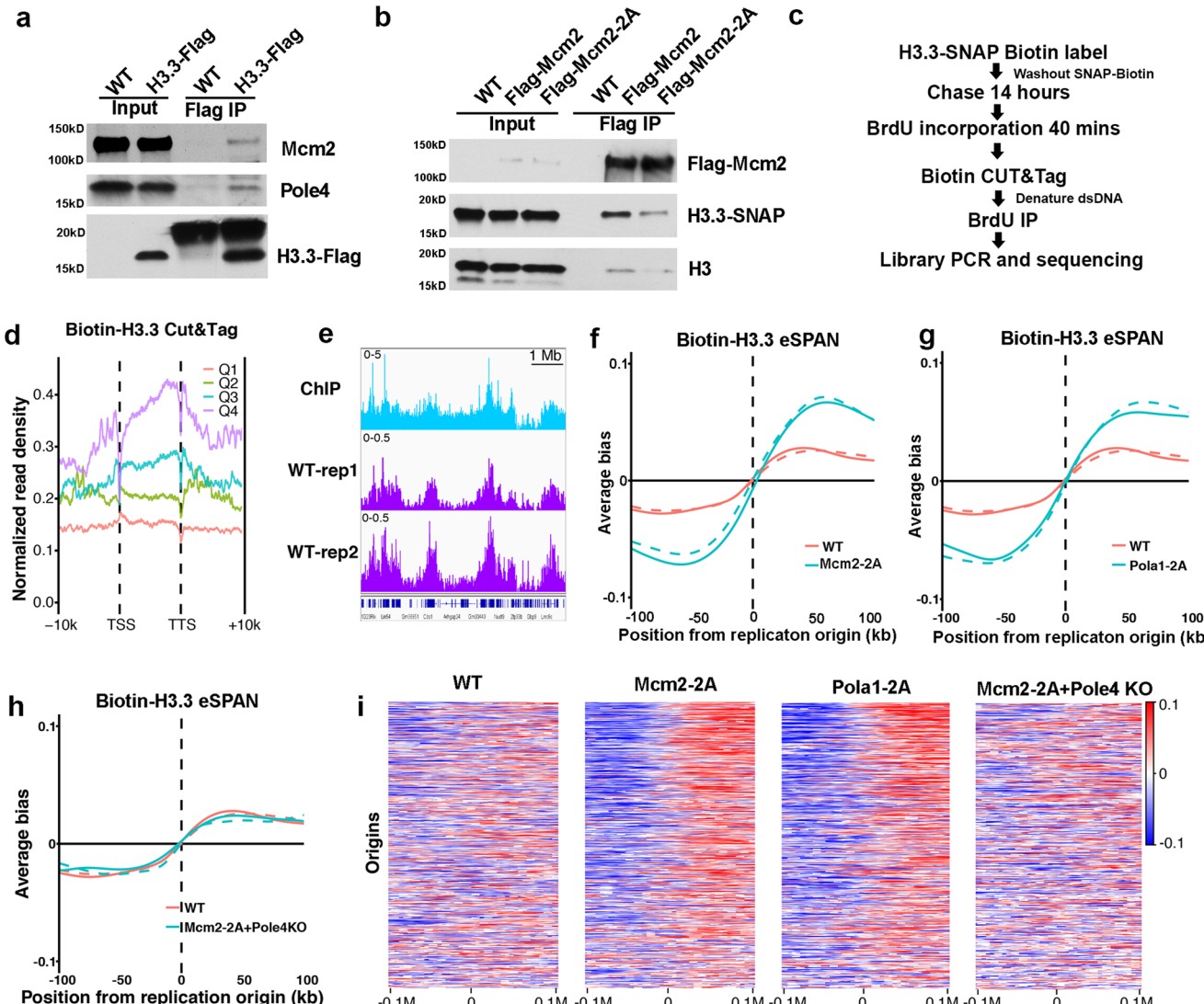

**Fig. 5 Mcm2 and Pole4 interact with H3.3 and facilitate parental H3.3 transfer to different replicating DNA strands. a** Both Mcm2 and Pole4 interact with H3.3 in vivo. WT or H3.3-Flag-tagged mES cells were collected for immunoprecipitation using anti-Flag antibodies. Proteins in the input extracts and IP samples were analyzed by Western blotting using Flag, Mcm2 and Pole4 antibodies. One representative result from three independent replicates was shown. **b** Mcm2-2A mutation reduces the Mcm2-H3.3 interaction. Cell extracts from WT, Mcm2-Flag, Mcm2-2A-Flag mouse cells containing H3.3-SNAP were used for immunoprecipitation using anti-Flag antibodies. Proteins in the input and IP samples were analyzed by Western blot using Flag, SNAP and H3 antibodies. One representative results from three independent replicates was shown. **c** An outline of the experimental procedures to analyze the distribution of parental H3.3-SNAP at replicating DNA using eSPAN. **d** Parental H3.3-SNAP CUT&Tag density at TSS and TTS at genes with different expression. Genes were separated into 4 groups based on their expression in mouse ES cells (Q1 = lowest, Q4 = highest). H3.3-SNAP CUT&Tag density was calculated for each group. **e** Snapshots of H3.3 ChIP-Seq (GSM2080326) and two repeats of H3.3-SNAP CUT&Tag density at selected region (chr5:99,655,789–106,254,109) in wild-type cells. The signals represent the normalized read count per million reads for each of the three indicated setting. **f** The average bias of parental H3.3-eSPAN peaks at 1548 replication origins in WT and Mcm2-2A mES cells. The eSPAN bias at each origin was calculated using the formula (W − C)/(W + C); W and C: sequence reads of the Watson strand and Crick strand, respectively. Two repeats are shown. **g, h** The average bias of parental H3.3-eSPAN peaks at 1548 replication origins in WT and Pola1-2A mES cells (**g**) as well as WT and Mcm2-2A Pole4 KO double mutant mES cells (**h**). Two repeats are shown. **i** Heatmaps of parental H3.3-SNAP eSPAN bias in WT, Mcm2-2A, Pola1-2A and Mcm2-2A + Pole4 KO mouse ES cells at each of the 1548 initiation zones, ranked from the most efficient (top) to the least efficient (bottom) ones based on OK-seq bias.

strands is exchanged with free H3.3. Nonetheless, these results provide additional experimental evidence supporting the idea that Mcm2-Pola1 and Pole3–Pole4 facilitate parental H3.3 transfer to lagging and leading strands, respectively, to ensure the faithful recycling of parental H3.3.

**Parental H3.3 proteins are likely distributed to the same chromatin regions following DNA replication.** The experiments presented above indicate that parental H3.3 is recycled via similar

mechanisms as H3.1 following DNA replication. To test this idea further, we first labeled H3.3-SNAP with the SNAP-biotin substrates, and then analyzed the distribution of H3.3-SNAP at 0, 5, and 11 h after release into fresh media without the SNAP-biotin substrates using CUT&RUN[41]. To quantify the dilution of parental H3.3-SNAP during this time course, we spiked in HeLa cells expressing H3.3-SNAP tag before performing CUT&RUN. We found that when normalized against total reads aligned to mouse genome, H3.3-SNAP CUT&RUN density was similar at all different time point. Moreover, the genome wide distribution of

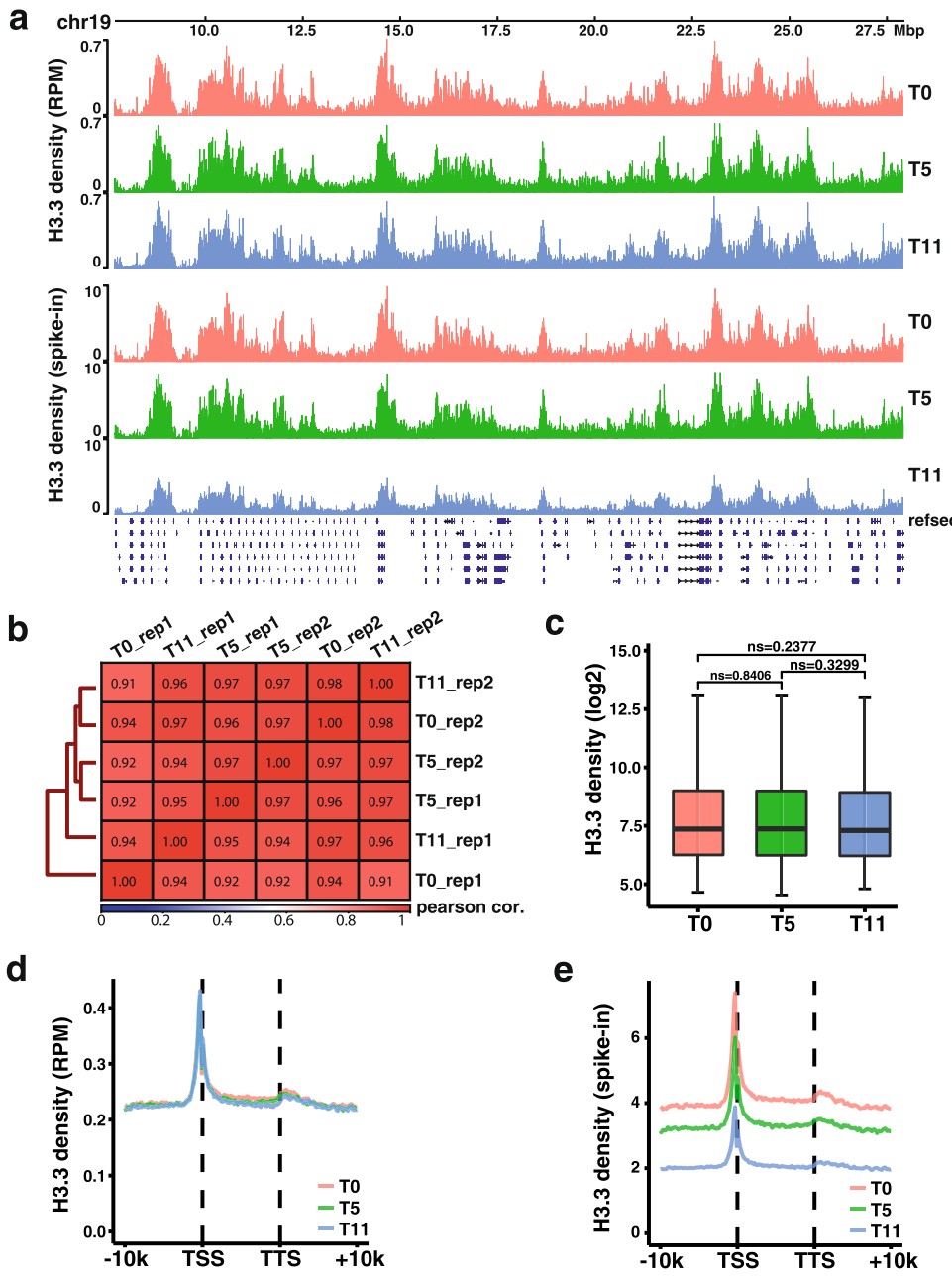

**Fig. 6 Parental H3.3 proteins are likely recycled at chromatin regions locally. a** Snapshot of Parental H3.3-SNAP CUT&RUN density at chromosome 19 (Chr19) at different three time points, 0 h (T0), 5 h (T5), and 11 h (T11) after release into fresh media without SNAP-biotin substrate in WT mouse ES cells. Reads were normalized either against total mapped reads at each time (top panels) or DNA from spike-in human HeLa cells at each time point (lower panels). **b** Pairwise Pearson correlation matrix of two biological replicates of parental H3.3-SNAP CUT&RUN signals at each time point across the whole genome using a 50 kb window. **c** The average of parental H3.3-SNAP CUT&RUN density at different time points (T0, T5, and T11) on chromatin regions enriched with H3.3 after normalizing to total mapped reads at each time point. Average of two biological replicates is shown. $n = 2461$ chromatin regions were analyzed for each time points. The center line, the box limits and the whiskers are defined as in Fig. 2c. Statistical analysis was performed by two-tailed unpaired Student $t$ test. The $P$ values are marked on the graphs (ns, no significant difference). **d** The average of H3.3-SNAP CUT&RUN density of each time point (T0, T5, and T11) at the TSS and TTS of 5271 genes localized at replicated chromatin regions after normalized with total mapped reads. Average of two biological replicates is shown. **e** H3.3-SNAP CUT&RUN density of each time point (T0, T5, and T11) at the TSS and TTS of 5271 genes localized at replicating chromatin when normalized against DNA from spike-in HeLa cells. The average of two biological replicates is shown.

parental H3.3-SNAP signals at different time point was highly correlated with each other (Fig. 6a, b). These results suggest that most parental H3.3 proteins are likely reassembled into nucleosomes locally following DNA replication. To test this idea further, we identified chromatin regions enriched with H3.3 using SCICER2[42] and calculated the average H3.3-SNAP CUT&RUN density at these regions. We found that H3.3 density at H3.3 peak

regions at different time points did not show significant differences (Fig. 6c). Similar results were obtained when analyzing the H3.3 CUT&RUN densities of replicated genes from transcription start site (TSS) to transcription termination site (TTS) at these different time points (Fig. 6d). In contrast, when normalized against spike-in, the H3.3 density was reduced with the most reduction of H3.3-SNAP at 11 h after release (Fig. 6a, e). Because

these CUT&RUN experiments were performed using asynchronized mouse ES cells, we could not deduce whether all parental H3.3-SNAP proteins were recycled following DNA replication. Nonetheless, these results are consistent with the idea that parental H3.3-SNAP proteins are reassembled into nucleosomes, likely locally like H3.1, by Mcm2-Pola1 and Pole3–Pole4 following DNA replication.

**Abnormal mitotic events are observed in cells with defects in parental histone transfer**. It has been reported that H3.3 KO leads to severe mitosis defects[43]. In mammalian cells, two genes, *H3f3a* and *H3f3b*, encode H3.3. Consistent with published results, we observed that deletion of either H3f3a or H3f3b resulted in a marked increase in mitotic defects, such as lagging chromosomes and anaphase bridges (Fig. 7a, b, Supplementary Fig. 8a). Next, we analyzed live-cell images collected in Fig. 3 and Supplementary Fig. S4 and 5, and observed that a dramatic increase in the percentage of cells with mitotic defects in Pole3 KO, Pole4 KO, and Mcm2-2A lines (Fig. 7c, d). Moreover, double mutants (Mcm2-2A/Pole3 KO and Mcm2-2A/Pole4 KO) cell lines displayed mitotic defects even more frequently than single mutant cell lines. Interestingly, Hira KO cells did not show defects in mitosis compared to the wild type, whereas Daxx KO showed a slight but statistically significant increase in the percentage of cells with mitosis defects (Fig. 7c, d). These results suggest that defects in parental histone recycling, including H3.3, by these replisome histone chaperones, contribute to defects in chromosome segregation. We also found an increase in G2/M phase cells in each of these mutant cells, including Pole3 KO, Pole4 KO, Mcm2-2A, and double mutants (Mcm2-2A/Pole3 KO and Mcm2-2A/Pole4 KO), compared to wild-type cells, consistent with mitotic defects (Fig. 7e; Supplementary Fig. 8f).

To test whether the mitotic defects observed in Mcm2-2A and Pole3 KO cells are associated with changes of expression of genes involved in mitosis, we analyzed gene expression in wild type, Mcm2-2A and Pole3 KO cell lines using RNA sequencing (RNA-seq). We identified 449 downregulated and 549 upregulated genes in Mcm2-2A cells, and 964 downregulated and 433 upregulated genes in Pole3 KO cells. Gene ontology analysis of the downregulated genes revealed that the expression of genes involved in mitotic cell division was affected in both Mcm2-2A and Pole3 KO cells, although these pathways were not in the top 20 pathways (Supplementary Fig. 8b–e). Several downregulated genes (*lgf1r*, *Top2b*, *Bcl2*, *PTTG1*, *App*, and *Pim1*) have been linked to chromosome segregation (Supplementary Tables 2 and 3)[44–49]. All together, these results suggest that faithful recycling of parental histones, including H3.3, is required for maintenance of genomic integrity during cell division.

## Discussion

Following DNA replication, replicating DNA is assembled into nucleosomes using both parental and newly synthesized histones. It is known that newly synthesized H3.1 and H3.3 are assembled into nucleosomes by different histone chaperones. However, due to the challenges inherent to tracking parental histones during DNA replication, there have been limited studies about the recycling of parental histones H3.1 and H3.3. Using live-cell imaging, we show that both H3.1 and H3.3 are largely recycled into daughter cells following mitotic cell division. Moreover, Mcm2-Pola1, and Pole3–Pole4 are involved in the transfer of both parental H3.1 and H3.3 to replication forks. These results provide mechanistic insight into recycling of parental histones H3–H4, a process that is critically important for the re-establishment of the epigenetic landscape following cell division.

**Is parental H33 stably inherited during S phase?** H3.3-containing nucleosomes account for about 20% of all nucleosomes in dividing cells and are distributed to actively transcribed regions as well as heterochromatin regions[9,10,19,38]. Various studies utilizing different systems and assays have provided conflicting conclusions as to whether H3.3-containing nucleosomes are inherited during mitotic cell divisions. An early study indicated that memory of active gene states can last through 24 cell divisions in the absence of transcription and that this memory is mediated by the levels of H3.3 at gene promoters[50]. Recently, it has been shown that budding yeast parental H3, which is structurally similar to H3.3 in higher eukaryotic cells, can remember their position along genomic DNA following DNA replication and gene transcription[27]. These studies indicate that H3.3-containing nucleosomes are likely inherited into daughter cells following cell divisions. By monitoring parental H3.1 using ChIP assays in mouse ES cells, it has been shown that H3.1 at silent chromatin regions are inherited locally following cell division, while H3.1 at actively transcribed regions, which are enriched with H3.3, are dispersed following cell division[28]. However, this study did not examine the distribution of parental H3.3 following cell division. In another study, through the monitoring of parental H3.1 and H3.3 in fixed cells using the SNAP tag proteins at different time points, it has been reported that parental H3.1 and H3.3 signals are lost over time and that this loss cannot be explained by the dilution during S phase, with a bigger loss of H3.3 than H3.1[37]. These results hint that not all H3.1 and H3.3-containing nucleosomes are stably inherited over time. In this study, we used live-cell imaging to measure the total levels of H3.3 and H3.1 using SNAP-tagged H3.1 and H3.3 expressed from their corresponding endogenous promoter. We found that the level of H3.1-SNAP or H3.3-SNAP at the G2 phase of the cell cycle is the same as that of G1/S phase, suggesting that both H3.1 and H3.3 are transferred to replicating DNA following DNA replication. Supporting this idea, H3.3-SNAP eSPAN experiments show that parental H3.3 are transferred to replicating DNA strands using similar factors for modified forms of H3. Moreover, we observed that H3.1-SNAP and H3.3-SNAP signals in the two daughter cells are about half that of their mother cell at G1, indicating that the daughter cells receive an equal amount of H3.1 and H3.3 following cell division. These results provide strong evidence that H3.3, much like H3.1, are largely stably segregated into daughter cells following DNA replication.

The observation of the stable recycling of parental H3.3 appears to be inconsistent with the loss of H3.3 over time observed previously[37]. There are two probable explanations to reconcile these seemingly contradictory findings. First, in our study, we monitored the H3.3-SNAP signals at individual cells and their two daughter cells during one cell division. In contrast, Clément et al. monitored the average H3.3-SNAP signals of many cells over time. Therefore, it is likely that cells at different stages of the cell cycle were used in the quantification. Second, fixed cells with pre-extraction were used for analysis[37]. The pre-extraction step may remove some parental H3.3 not tightly associated with chromatin. Our study monitored overall levels of parental H3.1 and H3.3 during one cell division using living cell image. This approach, however, cannot exclude the possibility that H3.3 is lost locally during gene transcription, an effect too small to be detected by live-cell imaging. Indeed, we noticed that the sum of H3.1- and H3.3-SNAP signals in two daughter cells at G1 are less than those in corresponding mother cells at G1/S in wild-type cells. While it is likely that the reduction is due to photobleaching overtime, it is equally possible that the reduction reflects loss of parental histone signals via histone exchange observed before[37]. Nonetheless, as we could not detect difference in H3.1-SNAP and H3.3-SNAP signals between G2 and G1/S in wild-type cells, we

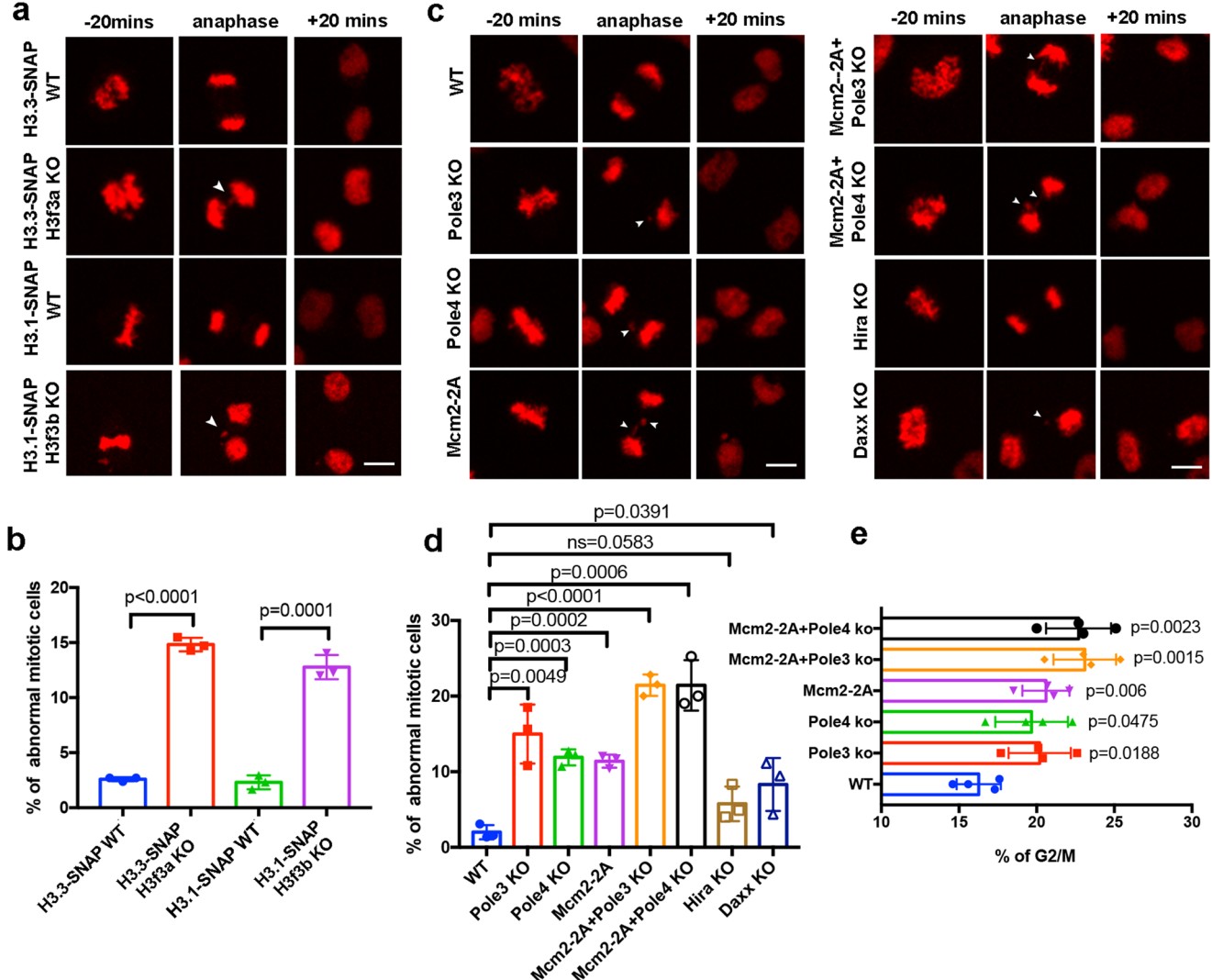

**Fig. 7 Mcm2-2A, Pole3 KO, and Pole4 KO mutants defective in parental histone transfer show abnormal chromosome segregation. a** Deletion of either H3f3a or H3f3b results in defects in mitosis. Representative live-cell images of H3.1/H3.3-SNAP at the indicated time points (20 min before (−) and after (+) anaphase) in WT, H3f3a KO and H3f3b KO mES cells (n > 160). Nuclear abnormalities marked by arrows including chromosome bridges, misaligned chromosomes and lagging chromosomes observed in anaphase was counted in H3f3a KO/H3f3b KO mES cells. Scale bar, 10 μm. **b** Percentages of abnormal mitotic cells in WT, H3f3a KO and H3f3b KO mES cells. Images at anaphase were used for quantification. **c** Representative live-cell images of H3.3-SNAP signals collected at indicated time (20 min before (−) and after (+) anaphase) in WT, Pole3 KO, Pole4 KO, Mcm2-2A, Mcm2-2A + Pole3 KO, Mcm2-2A + Pole4 KO, Hira KO, and Daxx KO mES cells (n > 170). The nuclear abnormalities mentioned in (**a**) were observed in Pole3 KO, Pole4 KO, Mcm2-2A, Mcm2-2A + Pole3 KO, Mcm2-2A + Pole4 KO, and Daxx KO mES cells. Defects are indicated by arrows. Scale bar, 10 μm. **d** Percentages of abnormal mitotic cells in WT, Pole3 KO, Pole4 KO, Mcm2-2A, Mcm2-2A + Pole3 KO, Mcm2-2A + Pole4 KO, Hira KO, and Daxx KO cell lines. Images at anaphase were used for quantification. **b**, **d** Data are presented as means ± SD. Statistical analysis was performed by two-tailed unpaired Student t test with the P values shown on the graphs (ns, no significant difference). **a**–**d** n > 160 cells from three independent experiments were analyzed. **e** The percentage of G2/M phase cells increases in WT, Pole3 KO, Pole4 KO, Mcm2-2A, single, Mcm2-2A + Pole3 KO, and Mcm2-2A Pole4 KO double mutant cell lines. Cells were collected for flow cytometry analysis of DNA content, and the percentage of cells at G2/M was calculated. Data are presented as means ± SD. Statistical analysis was performed by two-tailed unpaired Student t test, and the P values were marked on the graphs (ns no significant difference). Four independent replicates were conducted for each cell line.

suggest that almost all H3.3 and H3.1 are recycled following DNA replication. Consistent with this idea, we found that the chromatin distribution of parental H3.3-SNAP do not change following DNA replication (Fig. 6).

**Mcm2-Pola1 and Pole3–Pole4 are involved in the transfer of parental H3.3 during the S phase of the cell cycle.** Newly synthesized H3.1 and H3.3 are assembled into nucleosomes with the aid of different histone chaperones. However, it is not clear whether parental H3.1 and H3.3 are transferred to replicating

DNA with the help of different histone chaperones. By monitoring modifications on parental H3 and H4, we and others have shown that Mcm2-Pola1 and Pole4 are involved in the transfer of parental H3–H4 to replicating DNA[29–32]. Here, we show that these factors are also involved in the transfer of H3.3 behind replication forks. First, using live-cell imaging, we observed that the overall levels of H3.3 at the G2 phase of the cell cycle were reduced compared to G1 in each mutant cells tested, suggesting that these proteins are involved in the transfer of parental H3–H4 onto replicating DNA. Second, by measuring the relative amount of parental H3.3-SNAP at leading and lagging

strands of DNA replication forks with eSPAN, we observed that the transfer of parental H3.3-SNAP to leading or lagging strands is defective in each of the mutant cells. Together, these results indicate that the two pathways (Mcm2-Pola1 and Pole3–Pole4) involved in parental histone transfer defined using modifications on H3 and H4 are also important for recycling of parental H3.3 following DNA replication. Supporting this idea, we and others observed that Mcm2, Pole3–Pole4 bind to H3.3 in vitro and in vivo.

We observed that the effect of the Mcm2-2A mutation on the bias of H3K36me3 eSPAN peaks is larger than that of H3.3-SNAP-biotin eSPAN peaks. One possible explanation is that the antibodies against biotin used for the H3.3-SNAP eSPAN may not be accessible to all H3.3-SNAP-biotin proteins on chromatin, therefore minimizing the effects of Mcm2-2A mutation on the distribution of H3.3-SNAP at replicating DNA. In addition, low levels of endogenous biotin present in cells are recognized by biotin antibody, which may have impacted the fidelity of DNA generated through biotin-H3.3 CUT&Tag and thus generate non-specific signals. Consistent with this idea, we failed to detect any effects of the Mcm2-2A mutation on the distribution of H3.1-SNAP tag at replicating DNA. Alternatively, parental H3.3-SNAP, once transferred onto replicating DNA chromatin, exchanges with soluble H3.3 at a rate faster than that of H3K36me3 with soluble H3, which in turn minimizes the effect of the Mcm2-2A mutation on the distribution of H3.3-SNAP at replicating DNA strands. Nonetheless, our results support the idea that Mcm2, Pola1, Pole3, and Pole4 also function in the transfer of parental H3.3 to replicating DNA strands in mammalian cells.

In addition to Mcm2, Pola1, Pole3, and Pole4, we have shown that deletion of Daxx, a H3.3 chaperone that deposits newly synthesized H3.3 at repetitive heterochromatic regions, also reduces the transfer of parental H3.3 onto replicating DNA. Previously, it has been shown that Hira, another chaperone for newly synthesized H3.3, is also important for parental H3.3 transfer during transcription[36]. Therefore, it is likely that multiple chaperones are involved in recycling H3.3. Future studies are needed to address the interplay among these chaperones in recycling parental H3.3 during DNA replication and gene transcription.

**The faithful transfer of parental histones is important for genomic stability**. We observed that Mcm2-2A, Pole3, and Pole4 mutant cells defective in parental histone transfer also display abnormal chromosome segregation. These chromosome segregation defects are likely linked to the following three possibilities in these mutant cells. First, Mcm2-2A, and Pole3 and Pole4 KO cells may affect the expression of genes important for chromosome segregation. Second, the mitotic defects perhaps originate from the compromised transfer of parental histone H3 including H3.3. It has been shown that H3.3 depletion in murine cells, while having little effect on gene expression programs, alters heterochromatin structures at telomeres, centromeres and pericentric regions, thereby leading to mitotic defects[43]. We observed that deletion of either of the two genes encoding H3.3 (*H3f3a* and *H3f3b*) was sufficient to cause abnormal mitosis events (Fig. 7a, b). Interestingly, we did not observe significant mitotic defects in cells lacking Hira. Third, it is possible that the mitotic defects observed in these mutant cells is the manifestation of defects in transfer of all parental histone H3 and its variants including H3.1, H3.3, and CENPA. Consistent with this idea, it has been shown that Mcm2 also serves as a chaperone for parental CENPA[51–53]. In the future, it would be interesting to determine whether Pola1, Pole3, and Pole4 also function as a chaperone for parental CENPA. We have

shown previously that Mcm2-2A mutant cells defective in parental histone transfer also show defects in silencing of a handful of endogenous retrovirus elements[31]. In budding yeast, the Mcm2-3A mutant cells with defects in parental histone transfer showed an increased rate in loss of heterochromatin silencing[29]. Thus, the faithful transfer of parental histones H3–H4 is critical for both genome and epigenome integrity.

## Methods

**Cell culture and antibodies**. The mouse E14 ES cell line was kindly provided by Dr. Tom Fazzio (University of Massachusetts Medical School) and tested negative for mycoplasma. Cells were grown at 37 °C in DMEM (Corning) medium supplemented with 15% (v/v) fetal bovine serum, 1% penicillin/streptomycin (Invitrogen), 1 mM sodium pyruvate (Cellgro), 2 mM L-glutamine (Cellgro), 1% MEM non-essential amino acids (Invitrogen), 55 μM β-Mercaptoethanol (Sigma), and 10 ng/mL mouse leukemia inhibitory factor (mLIF) on gelatin-coated dishes in the presence of 5% $CO_2$ atmosphere.

Antibodies used in this study were as follows: anti-SNAP (P9310S, New England Biolabs, 1:1000 for WB), anti-Tubulin (12G10, DSHB, 1:3000 for WB), anti-Pole3 (A6469, Abclonal, 1:500 for WB), anti-Pole4 (A9882, Abclonal, 1:1000 for WB), anti-Daxx (#4533, Cell Signaling, 1:1000 for WB), anti-Hira (04-1488, Millipore Sigma, 1:1000 for WB), anti-H3.3 (C15210011, Diagenode, 1:1000 for WB), anti-Flag (F1804, SigmaAldrich, 1:2000 for WB), anti-His (CLH101AP, Cedarlane, 1:1000 for WB), anti-Mcm2 (ab4461, Abcam, 1:2000 for WB), anti-Biotin (A150-109A, Bethyl, 1:800 for CUT&Tag and CUT&RUN), antiBrdU (555627, BD Biosciences, 1:2000 for IP), anti-H3 (generated by immunizing rabbits with a synthetic peptide, MC1906, Cocalico Biologicals, 1:3000 for WB). anti-H3K36me3(61021, Active Motif, 1:200 for CUT&Tag). Uncropped scans of all Western blot results were included in the source data file.

**CRISPR-mediated gene editing and FastFUCCI cell lines generation**. CRISPR-Cas9 gene editing was performed following the published protocol[54]. To insert SNAP tag at the C-terminus of Hist1h3g or H3f3b, sgRNAs targeting the 3′ end of *Hist1h3g* or *H3f3b* sites were synthesized (IDT, Coralville, IA, USA) and cloned in to the PX459 vectors. Donor templates were generated by the insertion of the SNAP gene sequence flanked by the left homology arm and right homology arm sequence of the insertion site of *Hist1h3g* or *H3f3b* into the plasmid pBlueScript II SK. Targeting plasmids and donor plasmids were transfected into mES cells using Lipofectamine 3000 (Invitrogen). After selection with puromycin (2 μg/ml) for 2 days, single cells were seeded and grown into single cell clones. Clones were then picked, expanded and characterized by Sanger sequencing.

To generate Mcm2-2A mutant cell lines, sgRNA targeting Mcm2 sites were cloned into the PX459 vector, and single-stranded oligo DNA nucleotides (ssODNs) were synthesized (IDT, Coralville, IA, USA) as the repair template. Targeting plasmids and donor oligos were transfected into mESCs by using Lipofectamine 3000. Clones were identified using restriction enzyme digestion as the mutation introduced restriction site for the endonucleases BssHII and disrupted restriction site for AccI. Mutations were confirmed by Sanger sequencing.

Similar steps were followed to generate Pole3 KO and Pole4 KO cell lines, except that no ssODNs were needed for site-specific mutant. Depleted cells were identified by Surveyor nuclease assays and the knockout confirmed by Sanger sequencing.

pBOB-EF1-FastFUCCI-Puro was a gift from Kevin Brindle & Duncan Jodrell (Addgene plasmid # 86849)[39]. To generate FastFUCCI cell cycle reporter cell lines, the HEK 293T cell lines were transfected with pBOB-EF1-FastFUCCI-Puro and packaging plasmids. Supernatant from the culture medium containing virus was then collected and used to transduce WT and mutant mES cell lines. After selection with puromycin (2 μg/ml) for 2 days, single cells were seeded and grown into single cell clones. Clones were then picked, expanded and confirmed by fluorescence microscopy.

**Determination of mRNA half-life**. Cells were grown in 60% confluency and labeled with 4-sU (T4509, SigmaAldrich) at 150 μM final concentration for 20 min. Cells were then washed three times with fresh medium and incubated in medium with 2 mM Uridine (U3750, Sigma-Aldrich). Samples were collected at different time point (0, 2, 4, 6, and 8 h) and lysed at Trizol (15596026, Invitrogen). Total RNA was extracted following Trizol RNA extraction procedure. 60 μg total RNA of each sample were dissolved in 250 μl RNAase-free $H_2O$ and incubated at 65 °C for 10 min, followed by chilling on ice for 5 min. RNA was mixed with 50 μl biotinylation buffer (100 mM Tris-HCl, pH = 7.5, 10 mM EDTA, pH = 8.0), 100 μl DMF and 100 μl 1 mg/ml EZ-link HPDP Biotin (21341, Thermo Scientific) at 24 °C for 2 h. Biotinylated RNA was purified using phenol/chloroform RNA extraction and dissolved in RNAase-free $H_2O$. Samples were incubated at 65 °C for 10 min and chilled on ice for 5 min. 300 ug Streptavidin T1 beads (65601, Invitrogen) were added and rotated at 4 °C for 15 min. After extensive wash with wash buffer (100 mM Tris-HCl, pH = 7.5, 10 mM EDTA, pH = 8.0, 1 M NaCl and 0.1% Tween-20), nascent RNA was eluted in 100 mM DTT and were purified further

with Zymo RNA clean and purification kit (R1013, Zymo Research). 500 ng input total RNA and all the purified nascent RNA were used for RT with random hexamers (Invitrogen). Realtime quantitative PCR was performed in duplicates for each sample with SYBR Green PCR Master Mix on the CFX96 platform (BioRad Laboratories). Primers used were listed in Table S1.

**Acid extraction of histone**. Mouse ES cells were collected and washed once with PBS. Pellet was then washed with hypotonic lysis buffer (10 mM Tris-HCl, pH 8.0, 1 mM KCl, 1.5 mM MgCl2, 1 mM dithiothreitol (DTT) and protease inhibitor) and incubated in hypotonic buffer at 4 °C for 30 min. After washing three times with hypotonic buffer, nuclei pellet was resuspended in 400 μl 0.2 N HCl and rotated at 4 °C for 30 min. Supernatant was collected and mixed with 100 μl TCA, and incubated on ice for 30 min. After spin, histone pellet was washed twice with cold acetone, followed by air dry for 20 min. Proteins was dissolved in ddH$_2$O and supernatant was collected after spin and used for analysis by Western blot.

**Live-cell imaging**. Cells were incubated in labeling medium with either 1.5 μM SNAP-TMR or 1.5 μM SNAP-647-SiR (New England Biolabs) at 37 °C for 30 min. Cells were then washed three times with medium and incubated in fresh medium for 30 min followed by a replacement of fresh medium. Cells were then further incubated for at least 4 h, live-cell imaging was performed on a Nikon TiE Eclipse inverted microscope (Nikon, Inc., Melville, NY) equipped with a CSU-X1 spin-ning-disk unit (Yokogawa, Sugar Land, TX) and controlled with NIS Elements software (Nikon). Fluorescence was excited with a 561 nm (TMR) laser, or both 647 nm (647-SiR) laser and 488 nm (mAG-geminin) laser, and emission was collected through a standard rhodamine filter. Time-lapse frames were collected every 20 min for 16 h. At each time point, a z-series was collected at 3-μm focus intervals for a total of 21 μm of depth, and a maximum intensity projection was generated. Image J[55] was used to quantify fluorescence intensity at each indicated time point. At least n = 30 cells were quantified for each experiment. Error bars and P values were calculated from n cells scored in two independent experiments.

**H3.3-SNAP CUT&RUN analysis at different time point**. CUT&RUN was performed by following published protocol with some modifications[41]. H3.3-SNAP mES cells or H3.3-SNAP HeLa cells[22] were incubated in the labeling medium containing 1.5 μM SNAP-biotin (New England Biolabs) at 37 °C for 30 min. Cells were then washed three times with fresh medium and incubated in fresh medium for 30 min followed by a replacement with fresh medium. mES samples were then collected at 0 h, 5 h and 11 h after culturing in fresh medium, and mixed with 20% of SNAP-biotin labeled HeLa cells. Cells were washed twice with washing buffer (20 mM HEPES-NaOH pH 7.5, 150 mM NaCl, 0.5 mM Spermidine, 1× proteinase inhibitor cocktail) and immobilized to concanavalin A-coated magnetic beads. Cells were then incubated with biotin antibody (1:800) in antibody binding buffer (20 mM HEPES-NaOH pH = 7.5, 150 mM NaCl, 0.5 mM Spermidine, 2 mM EDTA, 0.04% Digitonin, 1× proteinase inhibitor cocktail) overnight at 4 °C. After washing with dig-wash buffer (0.04% Digitonin in washing buffer), cells were incubated with pre-assembled 2nd antibody + pA-MNase complex in dig-wash buffer for 1 h at 4 °C. After washing unbound pA-MNase, 2 mM CaCl2 was added to initiate digestion at 0 °C for 30 min. Reactions were stopped by mixing with 2XStop buffer (340 mM NaCl, 20 mM EDTA, 4 mM EGTA, 0.05% Digitonin, 25 μL 100 μg/ml RNase A) followed by incubation at 37 °C for 30 min. DNA in supernatant were purified using the QIAquick PCR Purification Kit (28104, Qiagen) and used for library preparation using the AccelNGS 1S Plus DNA library kit (Swift Bioscience, 10096). Each library DNAs were sequenced using pairedend sequencing by Illumina NextSeq 500 platforms at the Columbia University Genome Center.

**Analysis of parental H3.3-SNAP at replication forks using eSPAN and antibodies against Biotin**. Cells were incubated in the labeling medium with 1.5 μM of SNAP-Biotin (New England Biolabs) at 37 °C for 30 min, then washed three times with medium and incubated in fresh medium for 30 min. The media was replaced one more time before chasing for 14 h at 37 °C. BrdU (19-160, SigmaAldrich) was then added into the medium at 50 μM final concentration for 40 min. Cells were collected, washed twice with washing buffer (20 mM HEPES-NaOH pH 7.5, 150 mM NaCl, 0.5 mM Spermidine, 1× proteinase inhibitor cocktail) and immobilized to concanavalin A-coated magnetic beads. Cells were then incubated with biotin antibody (1:800) in antibody binding buffer (20 mM HEPES-NaOH pH = 7.5, 150 mM NaCl, 0.5 mM Spermidine, 2 mM EDTA, 0.1% BSA, 0.04% Digitonin, 1× proteinase inhibitor cocktail) overnight at 4 °C. After washing with dig-wash buffer (0.04% Digitonin in washing buffer), cells were incubated with pre-assembled 2nd antibody-pA-TN5 complex in dig-300 buffer (20 mM HEPES-NaOH pH 7.5, 300 mM NaCl, 0.5 mM Spermidine, 0.04% Digitonin, proteinase inhibitor cocktail) for 1 h at RT. After washing unbound pA-TN5, 10 mM MgCl2 was added to initiate tagmentation at 37 °C for 1 h. Reactions were stopped by mixing in 20 mM EDTA, 0.1% SDS and 0.1 mg/mL Proteinase K followed by gentle shaking at 37 °C overnight. Supernatants were purified using the QIAquick PCR Purification Kit (28104, Qiagen). The eluted DNA was subjected to an oligo-replacement reaction as detailed in[56]. Briefly, DNA samples were mixed with 0.5 mM dNTP mix, 0.5 μM mosaic end adaptor B and 1× Ampligase buffer under annealing program (50 °C, 1 min; 45 °C, 10 min; ramp to 37 °C at 0.1 °C/s and

hold), followed by adding T4 DNA polymerase and Ampligase and incubated at 37 °C for 30 min. The reaction products (5%) were saved as CUT&Tag samples, and the rest of the DNA was boiled at 98 °C for 5 min and chilled on ice. Denatured DNA were diluted in 1 ml BrdU IP buffer (1× PBS, 0.0625% Triton X-100) and mixed with BrdU antibodies (555627, BD Biosciences) and E. coli tRNA (Roche) at 4 °C for 2 h. Protein G beads (17-0618-02, GE healthcare) were then added and rotated at 4 °C for 1 h. After extensive wash, beads were incubated with elution buffer (50 mM Tris-HCl pH 8.0, 10 mM EDTA, 1% SDS) at 65 °C for 15 min with shaking. Supernatants were purified with ChIP DNA concentrator columns (D5205, ZYMO Research). Library PCR was performed using standard Illumina Nextera Dual Indexing primers. Libraries were sequenced using paired-end sequencing on Illumina NextSeq 500 platforms at the Columbia University Genome Center.

**Immunoprecipitation**. To immunoprecipitate proteins tagged with the Flag epitope, 3 × 10$^6$ cells were collected and suspended in 1 mL IP lysis buffer (50 mM HEPES- KOH, pH 7.4, 150 mM NaCl, 1% NP40, 10% glycerol, 1 mM EDTA and proteinase inhibitor). Lysates were dounced for 30 times with a tissue grinder (D8938-1SET; Millipore Sigma) and incubated on ice for 20 min. After clearing the lysate by centrifugation for 15 min, supernatants were incubated with 20 μl Flag M2 beads (A2220, SigmaAldrich) overnight at 4 °C. Beads were then extensively washed with IP lysis buffer. Proteins were eluted with 1× SDS sample buffer and analyzed by Western blot.

**Cell cycle analysis**. Exponentially growing mouse ES cells were collected and fixed in 70% ethanol in PBS, then rotated at 4 °C overnight. Cells were washed once with PBS and incubated in PI staining solution (0.1% Triton X-100, 0.2 mg/ml RNaseA, 0.02 mg/ml propidium iodide, 1× PBS) for 30 min at room temperature. Cells were analyzed by BD FACSDiva software of LSR II flow cytometer (BD Biosciences). Data were analyzed by Flowjo (version 10.7.1). Gating strategies are shown in Supplementary Fig. 9.

**Annexin V apoptosis detection**. 1 × 10$^5$ mES cells was collected for each sample and washed once in cold PBS. Cells were resuspended in 100 μl annexin-binding buffer (10 mM HEPES, 140 mM NaCl, and 2.5 mM CaCl2, pH 7.4), mixed with 5 μl aanexin V conjugate (A35122, Invitrogen) and 0.02 mg/ml propidium iodide. After incubation at RT for 15 min, 400 μl annexin-binding buffer was added. Stained samples were analyzed by Attune NxT software of Attune flow cytometer (Thermo Fisher Scientific). Data were analyzed by FCS Express (version 7). Gating strategy is shown in Supplementary Fig. 9.

**CUT&Tag, CUT&RUN, and eSPAN analysis**. For CUT&Tag analysis, the paired-end raw reads were trimmed to remove sequencing adaptors and low-quality reads using Trim Galore (version 0.6.7) (Developed by Felix Krueger at the Babraham Institute) with default parameters, and aligned to mouse (mm10) reference genome using Bowtie 2 (version 2.2.4)[57] with -no-mixed -no-discordant -no-dovetail -no-contain -local parameters. Multi-mapped reads were filtered using SAMtools (version 1.11)[58]. Picard Tools (version 2.23.8)[59] was used to remove ENCODE blacklisted regions and duplicate reads. Only paired-end reads with both ends mapped correctly were selected for further analysis. Genome coverage maps in bigwig format were calculated using deepTools bamCoverage (version 3.2.1)[60] and normalized to library size (reads-per-million; RPM). All heatmaps were plotted using deepTools (version 3.2.1). As there were non-specific bindings of Tn5 at open chromatin in CUT&Tag, therefore, we removed H3.3-SNAP CUT&Tag signals at open chromatin region based on published ATAC-seq in mES cell line (GSM3109355) for further analysis.

For eSPAN analysis, the consistent paired-end reads mapped to the Watson (W) and Crick (C) strands of the reference genome were separated by BEDTools bamtobed function[61] and in-house Perl programs. The bias of each bin was computed from these separated Watson and Crick reads within 5 kb window size using the formula Bias = $(W - C)/(W + C)$ across the whole genome. The bins with <4 sequencing reads would be ignored. The lines and heatmap of bias were drown based on the defined origins in mouse ES cells[31]. The bias from eSPAN for different marker was normalized using corresponding BrdU-IP-ssSeq. The bias was smoothed by flanking five bins for further visualization.

To analyze CUT&RUN with spike-in HeLa cells, the same procedures were used to align clean fastq files to a human (hg19) reference genome downloaded from GENCODE for spike-in control data. A normalization factor was calculated using the formula: normalization scale factor = 1000,000/library size of human. Next, from the mm10 aligned bam files, "proper-paired" reads were extracted using SAMtools with the output piped into BEDTools, producing BED files of reads that have been normalized to the number of reads aligned to the hg19 genome. BedGraphs of these files were generated as intermediary files to facilitate generation of BigWig coverage maps using the bedGraphToBigWig tool from UCSC (version 4)[62]. To detect the potential differential pattern of H3.3 deposition during cell cycle, total 2461 H3.3 islands were called using SICER2 setting the cutoff of FDR = 0.001[42]. Differential binding analysis were performed using edgeR[63] based on the total mapped reads normalization method and fail to detect the significantly differential pattern under the cutoff of fold change = 1.5 and FDR = 0.05.

**RNA sequencing**. Total RNA from H3.1-SNAP-tagged WT, Mcm2-2A, and Pole3 KO cells was isolated using RNeasy Plus Mini kit (74136, Qiagen). RNA-seq libraries were prepared and deep sequencing were performed by the Columbia University Genome Center. Two or three replicates for each sample were sequenced.

**RNA-seq analysis**. The paired-end reads of WT and Mcm2-2A were downloaded from GSE142996. RNA-seq library preparation and deep sequencing for Pole3 KO samples were performed by the Columbia University Genome Center. After trimming adapt and low-quality reads using Trim Galore (version 0.6.7), sample reads were aligned to the mouse genome (GENCODE mm10 primary assembly) using STAR (version 2.7.6a)[64]. Transcript quantification was performed using featureCounts (version 2.0.1)[65] to assign the unique mapped reads to exonic regions of GENCODE's vM17 annotation version[66]. Differentially expressed genes were identified using the edgeR pipeline[63] based on the count matrix. The resulting *P* values were corrected for multiple testing with false discovery rate (FDR) correction. The cutoff of FDR and Fold Change for differentially expressed genes was 0.05 and 1.5 separately. GO term and KEGG pathway enrichment analysis were separately performed using the "enrichGO" and "enrichKEGG" function from the clusterProfiler package (version 3.18.0)[67].

**Statistical analyses**. Data are presented as means ± SD. Differences between groups were evaluated using two-tailed unpaired Student *t* test (noted in figure legends). Statistical analysis was performed in GraphPad Prism software (version 7). All tests were considered significant at $p < 0.05$. Statistical analyses for all sequencing datasets were performed in R software (version 3.6.3). Statistical parameters, and statistical methods used, error bar definitions and sample sizes were reported in the figures and corresponding figure legends. Where outliers were removed for plotting purposes, the removed data points were still used for statistical analyses.

**Reporting summary**. Further information on research design is available in the Nature Research Reporting Summary linked to this article.

## Data availability

The data that support this study are available from the corresponding author upon reasonable request. The raw and processed sequencing data generated in the course of this study are available in the Gene Expression Omnibus (GEO) database under accession code GSE183065. Previously published RNA-seq data under the accession code GSE142996 were also used in this study. The GTF and FASTA files used for Bioinformatics analysis (mm10, GENCODE release M27; hg19, GENCODE release 28) can be downloaded from GENCODE (https://www.gencodegenes.org). Source data are provided with this paper.

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

## Acknowledgements

We thank Albert Cardona Serra for critical reading of the paper, Emilia L. Munteanu and Theresa C. Swayne for helping with Confocal microscopy operation, which were performed in the Confocal and Specialized Microscopy. DNA sequencing was performed in Columbia Genome Center. Cell cycle analysis was performed in Flow Cytometry Core. This work is supported by NIH grants R35GM118015 (Z.Z.). These shared Resource of the Herbert Irving Comprehensive Cancer Center at Columbia University is supported by NIH/NCI Cancer Center Support Grant P30CA013696 (A.R.) and NIH grant S10RR027050 (R.C.).

## Author contributions

X.X. and Z.Z. conceived the project. X.X. and Z.L. performed experiments. S.D. performed the data analysis. X.H. provided RNA-seq datasets. R.H. helped paper editing. Z.Z. supervised the study. X.X., S.D. and Z.Z. wrote the paper with comments from all authors.

## Competing interests

The authors declare no competing interests.
