## [Peer Review File · Nature Communications]

REVIEWER COMMENTS

Reviewer #1 (Remarks to the Author):

In the present manuscript Xu, Duan et al., report the analysis of SNAP-tagged H3.1 and H3.3 transmission during DNA replication in mouse ES cells and the effect of the previously reported MCM2-2A and POLE3-4 histones binding mutants in this process. They proceed to confirm some of their findings using their previously developed eSPAN method.

Overall, the work is of interest for Nature Communication as this is an expanding area of research at the interplay between genome and epigenome stability. In addition, several laboratories have reported contrasting results using similar (e.g. SNAP-tag H3.1 and H3.3) or different methods. As such, alternative experimental approaches and work are welcome, if properly controlled.

Having said that, I have several points (major and minor) that I believe the authors have to address to deserve publication in Nature Communications.

- 1) Technical point. how were cells identified as G1/S ? Is this G1/S transition or mixed cells in G1 and S ? were cells previously synchronized ? if not, differences in cell cycle progression between WT, MCM2-2A and POLE3-POLE4 KO might account for some of the observed differences in the TMR signal. A control with synchronized cells should be done or differences in the percentage of G1 and S-phase cells (on the dynamics of the TMR signal) have to be excluded in a different manner. This is important as the authors observe very small differences in late G2 signal in POLE4 KO and MCM2 2A mutant cells compared to G1/S.
- 2) Minor point related to figures 2 and 3. Is the sum of daughter 1 + 2 TMR signal lower than G1/S mother (as I suppose) ? This should be described (with statistics).
- 3) Surprisingly the effect of the combined deletion of POLE3/4 and MCM2-2A is similar to deletion of the single histone chaperone motifs (on TMR signal at least). How do author explain this ? also, do double mutant cells have any proliferation or differentiation defects (given also their different transcriptional profile as the authors later show) ?
- 4) is mutation of the pola1 motive (previously described by the same authors) additional to mcm2 instead ? the use of this other model might reinforce the specificity of authors finding with the SNAP-tag system.
- 5) the finding described on HIRA and DAXX might be a major figure in the paper (but this is up to the authors)
- 6) what is the effect of combined POLE4 KO and MCM 2A mutations on the eSPAN for H3.3 ? as there is not overall difference on TMR values (between single and double mutants) it is important to report and discuss this data.
- 7) I am not sure what the authors suggest with the RNA Seq data (that overall are poorly presented in Sup. Figure 6). Mutant cells have a different cell cycle distribution and thus a different transcriptional profile. In addition to this, mitotic division pathway appears not the be the most significantly affected gene network. Do authors have alternative explanation for the observed genetic instability (e.g. telomeric disfunction, DNA damage accumulation during DNA replication or simply defective heterochromatin maintenance) ? could this be caused by defective recycling of CENPA (alternatively) ?

Reviewer #2 (Remarks to the Author):

Using SNAP-tagged system and eSPAN method combined with live single cell imaging, this manuscript reports that surprisingly the parent H3.3 histones, like H3.1, are faithfully recycled following DNA replication and stably transferred into daughter cells; Mcm2, Pole3 and Pole4 are involved in parent histone transfer for both H3.1 and H3.3; and Mcm2, Pole3, or Pole4 mutants cause mitotic defects. These findings are important and would improve our understanding of how epigenetic memory is faithfully transmitted as cells divide to maintain genomic integrity. There are a number of concerns that need to be addressed to support their claims and strengthen the manuscript.

Specific comments:

1. An unexpected finding of this study is that the amount of parental H3.1 at G1/S was the same as at G2 and each daughter cell received half of H3.1 from the mother cell following cell division. What even more surprising is that the similar result was seen for H3.3, although H3.3 has an estimated half-life of ~ 24 h. These findings are new and interesting but, as the authors also pointed out, are different from other groups' results. To justify the statement the authors need to include additional controls to show if H3.3-SNAP mRNA and protein have similar stabilities to that of endogenous H3.3. There is a possibility that SNAP fusion and/or possible changes in 3' mRNA structure due to gene editing prevent H3.1/H3.3-SNAP mRNA and protein from being degraded at the same rate as the endogenous respective histones, which may have contributed to the observed results. In addition, the percentage of H3.1/H3.3-SNAP in total H3 needs to be calculated, which could provide important information such as indirect evidence about its stability. Based on Supplementary Figure 1C, the amount of H3.1/H3.3-SNAP appears to be higher than expected ($\sim 10\%$ of total?). If this is the case, the data implies H3.1/H3.3-SNAP may be more stable than endogenous H3.1 or H3.3.

2. TMR signals in each cell were measured to quantify the amount of H3.1/H3.3-SNAP at different time points. Based on these measurements, it was concluded that parental H3.1 and H3.3 are faithfully recycled following one cell division. However, stating that "...parental H3.1 proteins are stably reassembled into nucleosomes following DNA replication..." (page 7) may not be accurate, since overall it measures both chromatin-bound and free nuclear histones and the signals measured do not necessarily represent the histones that are already assembled into chromatin. And this might be one of the reasons why different results were obtained by Clement et al. (ref 37) where they removed free histones by fixation and compared chromatin-bound histones only. It would be worthwhile to perform H3.3-SNAP eSPAN as shown in Figure 4c,d but in "daughter" cells, and run comparison to see if the H3.3-SNAP level is about half of that seen in Figure 4d. This could provide evidence of whether parent H3.3-SNAP is stably reassembled into chromatin.

3. Mcm2 is an DNA helicase and one of key component for replication fork. It is possible that Mcm2 mutant induced defect in DNA replication first, which consequently resulted in the observed decrease in parent H3.1/H3.3-SNAP amount following DNA replication. Same question is for Pole3 and Pole4 mutant as well. How to distinguish between replication and chaperone functions of Mcm2, Pole3 and Pole4?

4. SNAP has molecular weight of ~ 20 kD. It may interfere with binding of H3.1/H3.3-SNAP to chromatin proteins and deregulate gene expression. This needs to be taken account when RNA-seq data is analyzed.

5. Abstract and Introduction: the conclusions were drawn from the experiments using SNAP-tagged system and it needs to be pointed out in the Abstract and Introduction.

6. Abstract: "same mechanisms" needs to be replaced, as while Mcm2, Pole3 and 4 are

shared by H3.1 and H3.3, we do not know yet if other chaperones are used by H3.1 and H3.3 differently.

7. Line 13 in page 3: "telomeric heterochromatin". H3.3 is also localized at other heterochromatin regions as well.

8. Figs 2, 3 and 5: double mutants caused more severe mitotic defects than single mutant, whereas no difference of H3.1/H3.3 recycling was seen between double and single mutant. Why?

9. Supplementary Figure 1D: are any chromosome aberrations were observed? How about cell cycle?

10. Supplementary Figure 5h. the lines and names in the legend are not aligned. Please correct.

Reviewer #3 (Remarks to the Author):

In the manuscript titled "Stable inheritance of H3.3-containing nucleosomes during mitotic cell divisions", Xiaowei Xu et al tracked the dynamic of parental histone H3.1 and H3.3 throughout one cell cycle using the live cell imaging. Overall their results showed that Mcm2 and Pole3/Pole4 regulate the recycling of histone H3.1 and H3.3 during DNA replication and maintain the genome integrity. The manuscript revealed the transfer of parental H3.1 and H3.3 is mediated by the same mechanism, which contributes to the epigenetic memory and the maintenance of genome stability. However, there are still some ambiguities in this manuscript, and some concerns need to be addressed before this manuscript is considered to publish.

1. The authors used the fluorescent staining to quantify the recycle of parental H3 during the DNA replication processes. However, how to rule out the fluorescent labelled histones, which are evicted from chromatin, are not assembled or deposited into chromatin during the DNA replication process? The authors may perform immunofluorescent staining, in which the free fluorescent histone can be washed away, to rule out the free fluorescent labelled histones.

2. The authors performed parental H3.3-SNAP CUT-Tag on nascent DNA, we wonder whether the distribution of parental H3.3 have specificity on the genome compared with total H3.3. Moreover, are the parental H3.3 immediately restored to the half of the level before replication or restored gradually? Are parental H3.3 well preserved at all peak regions during the cell cycle or only at some specific genomic regions? We recommend that the authors had better to perform time-scale (chase for different time after BrdU pulse) parental H3.3-SNAP CUT-Tag to analyze whether there are some specific patterns for the recycling of parental H3.3 during cell cycle, and find out which regions are regulated by Mcm2 or Pole3/Pole4.

3. This manuscript revealed a portion of H3.3 has to be recycled during DNA replication, what genes are regulated by these "replication-coupled H3.3"?

4. The conclusion "All together, these results suggest that faithful recycling of parental H3.3 is required for maintenance of genomic integrity during cell division." in page 13 is not accurate. Data in this manuscript cannot rule out the defects in the Mcm2 and Pole3/Pole4 KO cells is caused by the disturbed recycling of parental H3.1. So we recommend use the word "parental histone" instead of "parental H3.3".

5. For figure 5 C, we would recommend adding scale bar.

6. "Both McM2 and Pole4 coimmunoprecipitated with H3.3-Flag, confirming that that they interact with histone H3.3 in vivo (Fig. 4a)". Please notes : the "that" is redundant.

REVIEWER COMMENTS

Reviewer #1 (Remarks to the Author):

In the present manuscript Xu, Duan et al., report the analysis of SNAP-tagged H3.1 and H3.3 transmission during DNA replication in mouse ES cells and the effect of the previously reported MCM2-2A and POLE3-4 histones binding mutants in this process. They proceed to confirm some of their findings using their previously developed eSPAN method.

Overall, the work is of interest for Nature Communication as this is an expanding area of research at the interplay between genome and epigenome stability. In addition, several laboratories have reported contrasting results using similar (e.g. SNAP-tag H3.1 and H3.3) or different methods. As such, alternative experimental approaches and work are welcome, if properly controlled.

Having said that, I have several points (major and minor) that I believe the authors have to address to deserve publication in Nature Communications.

Response: We thank the reviewer for his/her time to review this exciting story and for his/her very positive comments.

Reviewer #1

1) Technical point. how were cells identified as G1/S? Is this G1/S transition or mixed cells in G1 and S? were cells previously synchronized? if not, differences in cell cycle progression between WT, MCM2-2A and POLE3-POLE4 KO might account for some of the observed differences in the TMR signal. A control with synchronized cells should be done or differences in the percentage of G1 and S-phase cells (on the dynamics of the TMR signal) have to be excluded in a different manner. This is important as the authors observe very small differences in late G2 signal in POLE4 KO and MCM2 2A mutant cells compared to G1/S.

Response: Thank the reviewer for suggestions. We did the followings to address the reviewer's concerns. First, we modified the text and described in more detail how we defined G1/S transitions (page 6-7). Previously, the G1/S transition of each cell was estimated in asynchronized ES cell populations. Briefly, based on the chromatin condensation revealed by H3.1-SNAP or H3.3-SNAP live cell imaging, we first identified a cell at the mitosis and used this time point as the reference point to choose cells having H3.1-SNAP and H3.3-SNAP images at least 11 hrs before mitosis and 1 hour after mitosis. We then define 11 hours before mitosis as G1/S, 1 hour before mitosis as G2, and 1 hour after mitosis as the next G1. To further clarify this point, we also modified figure labeling to increase clarity of definition of G1/S.

Second, we agreed with the reviewer that it is a great idea to analyze the intensity of H3.1-SNAP and H3.3-SNAP signals during cell cycle progression using synchronized cells. However, as the reviewer knows, it is almost impossible to synchronize mouse embryonic stem cells and release these cells unperturbed into the cell cycle. Therefore, to address the reviewer's concern experimentally and to test whether these mutations affected cell cycle progression, we decided to use the fluorescence ubiquitination-based cell cycle indicator (FUCCI). This system relies on two proteins, Cdt1 and its inhibitor, geminin that are involved in DNA replication control. The

expression of Cdt1 peaks in the G1 phase, whereas the expression of geminin peaks in S and G2 phase, but is low in late mitosis and G1 phase. The levels of these proteins are controlled by ubiquitin mediated degradation. Therefore, we expressed the cell cycle indicator mKO2-Cdt1, and mAG-Geminin in WT, Pole4 KO, Mcm2-2A, Mcm2-2A+Pole4 KO and Pola1-2A cells. During the subsequent and initial live cell image analysis, we realized that the fluorescent intensity of mKO2-Cdt1 signals were low in all cell lines. Therefore, to avoid photo bleach during 16hrs live cell imaging, we decided to use mAG-Geminin signals to mark cell cycle. Based on the appearance of mAG-Geminin signals, we estimated that the average time from G1/S to mitosis was 11 hrs for WT, Pole4 KO and Mcm2-2A single mutant cells, and 11.6 hrs for Mcm2-2A+Pole4 KO double mutant cells and Pola1-2A single mutant cells (new Fig. 4b). This result supports previous estimation that it takes wild type, Mcm2-2A and Pole4 KO cells 11 hrs to progress from G1/S to mitosis. Therefore, the original definition of G1/S in wild type and these mutant cells in Fig. 1, Fig. 2 and Fig. 3 held based on these results. Based on these results, we also re-analyzed the effects of Mcm2-2A+Pole4 KO by using images at 12 hours before mitosis as G1/S.

Finally, we also collected live cell image of H3.3-SNAP signals during cell cycle progression in cells expressing the FUCCI cell cycle indicator, and used the appearance of mAG-Geminin signals as G1/S for each WT, Pole4 KO, Mcm2-2A, Mcm2-2A+Pole4 KO and Pola1-2A cells. This independent analysis confirmed that these mutants affect recycling of parental histone H3.3 following DNA replication (new Fig. 4c-h).

Reviewer #1

2) Minor point related to figures 2 and 3. Is the sum of daughter 1 + 2 TMR signal lower than G1/S mother (as I suppose)? This should be described (with statistics).

Response: We followed the reviewer's suggestion and compared the H3.1-SNAP and H3.3-SNAP signals between G1/S and the SUM of the H3.1-SNAP and H3.3-SNAP signals in two daughter cells of the next G1 of wild type and each of the mutant cells. We found that the sum of H3.1-SNAP and H3.3-SNAP signals in two daughter cells of next G1 were slightly but statistically lower than the signals at G1/S transition in wild type cells (all 4 independent experiments with H3.1-SNAP and 6/8 of H3.3-SNAP). However, we did not observe significant changes in H3.1- or H3.3 SNAP signals between G2 and G1/S in any of these experiments of wild type cells. Therefore, to ascertain the effects of each mutant on the recycling of parental histones, we compared the H3.1-or H3.3-SNAP signals between G2 and G1/S in each mutant cells. In the revised manuscript, we described this result and potential reasons for the reduction of the sum of H3.1-SNAP and H3.3-SNAP signals in two daughter cells at G1 compared to those at G1/S of their corresponding mother cells in result (p8) and discussion (p22).

Reviewer #1

3) Surprisingly the effect of the combined deletion of POLE3/4 and MCM2-2A is similar to deletion of the single histone chaperone motifs (on TMR signal at least). How do author explain this? also, do double mutant cells have any proliferation or differentiation defects (given also their different transcriptional profile as the authors later show)?

Response: We thank the reviewer for pointing this out. One possible reason for this result is due to the sensitivity of our assays. To compound this issue, we observed that Mcm2-2A+Pole3 KO and Mcm2-2A+Pole4 KO double mutant cells showed significant more cell death than either single mutant alone (new Supplementary Fig. 3d). We speculate that double mutant cells with a dramatic reduction in parental histone recycling will undergo apoptosis and could not be detected by live cell imaging (see p10).

The double mutant cells show a slight proliferation defect compared to single mutant cells. In addition, we observed that it takes about 11.6 hours for Mcm2-2A+Pole 4 KO to progress from G1/S to mitosis, whereas it takes about 11 hours for wild type, Mcm2-2A and Pole 4 KO to progress from G1/S to mitosis (new Fig. 4b).

We also analyzed the effects of Pole4 KO, Mcm2-2A and Mcm2-2A+Pole4 KO mutation on differentiation using embryonic body (EB) formation assays, which mimics the development of three germ layers. We found that in general the double mutant dramatically affected the differentiation of mES cells compared to wild type or each of the single mutant analyzed based on analysis of the expression of pluripotent gene (Oct4) and chosen lineage specific genes representing each germ layer (see Letter Figure 1). Currently, we are trying to understand why these mutations affect the differentiation of ES cells.

Letter Figure 1: Mcm2-2A and Pole4 KO double mutant cells dramatically inhibited mES differentiation. RT-qPCR analysis of expression pluripotency genes (Oct4) and lineage specific genes (Gata4, Cdx2, Nestin and Brachyury) in WT, Pole4 KO, Mcm2-2A and Mcm2-2A+Pole4 KO mES cells during EB formation. X axis represents the number of days without LIF. Data are presented as means \pm SEM from three independent experiments. (* $P < 0.05$, ** $P < 0.01$, *** $P < 0.001$, Student's t-test)

Reviewer #1

4) is mutation of the *pola1* motive (previously described by the same authors) additional to *mcm2* instead? the use of this other model might reinforce the specificity of authors finding with the SNAP-tag system.

Response: We followed the reviewer's suggestions and analyzed the effects of Pola1-2A mutant, which is defective in binding to H3-H4, on the recycling of parental H3.3 using both live cell imaging and eSPAN. We observed that the recycling of parental H3.3 was defective in Pola1-2A cells based on live cell imaging analysis (new Fig. 4a,g,h). Moreover, we observed that H3.3-SNAP eSPAN signals in Pola1-2A mutant cells showed an increased bias towards leading strands compared to wild type cells (new Fig. 5g,i). These results indicate Pola1, like Mcm2, also facilitates parental H3.3 transfer to lagging strand.

Reviewer #1

5) the finding described on HIRA and DAXX might be a major figure in the paper (but this is up to the authors)

Response: We thank the reviewer for suggestions. After inclusion additional figure (Fig. 4) based on the experiments proposed by the reviewer, we decided to keep the effects of HIRA and DAXX mutations on parental histone transfer in supplemental Figures. In the revised manuscript, there are 7 main figures. However, if the reviewer insists on including these results into main figure, we are happy to do it.

Reviewer #1

6) what is the effect of combined POLE4 KO and MCM 2A mutations on the eSPAN for H3.3? as there is not overall difference on TMR values (between single and double mutants) it is important to report and discuss this data.

Response: We followed the reviewer's suggestion and performed both parental H3.3-SNAP eSPAN and H3K36me3 eSPAN in Mcm2-2A+Pole4 KO double mutant cells. We observed that H3.3-SNAP and H3K36me3 eSPAN peaks in double mutant cells showed similar bias as WT cells (new Fig. 5h, i). These results are consistent with the idea that the transfer of parental histones to lagging and leading strands is compromised in Mcm2-2A and Pole4 KO cells, respectively. Therefore, the effects of Mcm2-2A and Pole4 KO in double mutant cells will cancel each other. We have now described the results in main Fig. 5 and Supplemental Fig. 7.

Reviewer #1

7) I am not sure what the authors suggest with the RNA Seq data (that overall are poorly presented in Sup. Figure 6). Mutant cells have a different cell cycle distribution and thus a different transcriptional profile. In addition to this, mitotic division pathway appears not to be the most significantly affected gene network. Do authors have alternative explanation for the observed genetic instability (e.g. telomeric dysfunction, DNA damage accumulation during DNA replication or simply defective heterochromatin maintenance)? could this be caused by defective recycling of CENPA (alternatively)?

Response: We thank the reviewer's suggestions and comments. In the revised manuscript, we modified the text for clarifications. We agree with the reviewer that mitotic defects observed in these mutant cells are likely due to multiple reasons. In the revised manuscript, we tried our best to clarify this situation and proposed that there are three potential mechanisms for the observed mitotic defects in Mcm2-2A, Pole3 KO and Pole 4 KO cells (see discussion, p24-25). One involves in the recycling of parental CENPA, as the reviewer suggested. It has been reported that

CENPA inheritance relies on the interaction with Mcm2 protein [1-3]. We cited these studies in the discussion.

Reviewer #2 (Remarks to the Author):

Using SNAP-tagged system and eSPAN method combined with live single cell imaging, this manuscript reports that surprisingly the parent H3.3 histones, like H3.1, are faithfully recycled following DNA replication and stably transferred into daughter cells; Mcm2, Pole3 and Pole4 are involved in parent histone transfer for both H3.1 and H3.3; and Mcm2, Pole3, or Pole4 mutants cause mitotic defects. These findings are important and would improve our understanding of how epigenetic memory is faithfully transmitted as cells divide to maintain genomic integrity. There are a number of concerns that need to be addressed to support their claims and strengthen the manuscript.

Response: We thank the reviewer for his/her time to review this exciting story and for his positive comments.

Reviewer #2

Specific comments:

1. An unexpected finding of this study is that the amount of parental H3.1 at G1/S was the same as at G2 and each daughter cell received half of H3.1 from the mother cell following cell division. What even more surprising is that the similar result was seen for H3.3, although H3.3 has an estimated half-life of ~24 h. These findings are new and interesting but, as the authors also pointed out, are different from other groups' results. To justify the statement the authors need to include additional controls to show if H3.3-SNAP mRNA and protein have similar stabilities to that of endogenous H3.3. There is a possibility that SNAP fusion and/or possible changes in 3' mRNA structure due to gene editing prevent H3.1/H3.3-SNAP mRNA and protein from being degraded at the same rate as the endogenous respective histones, which may have contributed to the observed results.

Response: We thank reviewer's suggestions. We followed the reviewer's suggestions and evaluated whether SNAP tag altered the stability of H3.3 mRNA and protein using multiple approaches. First, we compared the level of endogenous H3f3b in E14 WT with H3f3b-SNAP in the tagged cell lines by RT-PCR. We did not observe significant changes in the levels of H3f3b mRNA in wild type and H3.3-SNAP tagged cells (new Supplementary Fig. 2a). Furthermore, we labeled nascent RNA using 4-sU and measured the nascent RNA decay rates in WT and H3f3b-SNAP by following published procedures [4]. We observed that the half-life of nascent RNA of H3f3b and H3f3b-SNAP was quite similar (5.7 vs 5.6 hours, new Supplementary Fig. 2b). Together, these results indicate SNAP tag does not change the mRNA stability of H3f3b.

Finally, we also compared the levels of H3.3-SNAP to H3.3 produced from *H3f3a* gene, another gene encoding H3.3 using Western blot (new Supplementary Fig. 2c,d) and estimated that the

ratio of H3f3a/H3f3b-SNAP was about 1.6 fold. Based on the RNA-seq, the ratio of H3f3b mRNA over H3f3a is about 1.7 fold. While it may be less efficient for transferring larger H3f3b-SNAP proteins than H3f3a proteins, these results suggest that it is unlikely that H3.3-SNAP tag will increase the stability of H3.3. If anything, it may reduce the stability.

Reviewer #2

2. TMR signals in each cell were measured to quantify the amount of H3.1/H3.3-SNAP at different time points. Based on these measurements, it was concluded that parental H3.1 and H3.3 are faithfully recycled following one cell division. However, stating that "...parental H3.1 proteins are stably reassembled into nucleosomes following DNA replication..." (page 7) may not be accurate, since overall it measures both chromatin-bound and free nuclear histones and the signals measured do not necessarily represent the histones that are already assembled into chromatin. And this might be one of the reasons why different results were obtained by Clement et al. (ref 37) where they removed free histones by fixation and compared chromatin-bound histones only.

Response: We share the concerns of the reviewer. While we measured the total H3.1 and total H3.3 in cells using integrated TMR signals, I would like to point out that in cells the majority of histone proteins is on chromatin. Moreover, we started the imaging at least 4 hours after TMR labeling, which will further reduce the amount of H3.3-SNAP labeled with TMR that are not assembled into chromatin. While it is possible that nucleosomal H3.1 and H3.3 can exchange with free H3.1 and H3.3 locally, it has been shown that this exchange is minimal during S phase of cell cycle [5]. To address this concern, we now remodified our sentence in *main text*, saying "*vast majority of parental H3.1 proteins are stably inherited following DNA replication*".

Reviewer #2

It would be worthwhile to perform H3.3-SNAP eSPAN as shown in Figure 4c,d but in "daughter" cells, and run comparison to see if the H3.3-SNAP level is about half of that seen in Figure 4d. This could provide evidence of whether parent H3.3-SNAP is stably reassembled into chromatin.

Response: The experiments proposed by the reviewer is interesting. However, it is almost impossible to do for the following two reasons. As stated above, it is impossible to synchronize mouse ES cells and isolate daughter cells at next G1 phase. Second, the eSPAN experiment relies on analysis of replicating chromatin regions. To address the reviewer's concern, we performed Biotin-H3.3-SNAP CUT&RUN at different time after labeling H3.3-SNAP with biotin (Fig. 6). When normalized against total mapped sequence reads, we found that H3.3-SNAP signals were unchanged from 0 hour to 11 hours after labeling. However, when normalized against DNA from spiked-in HeLa cells with H3.3-SNAP, we observed that there is significant reduction of H3.3-SNAP signals. These results are consistent with the idea that most of H3.3-SNAP are likely recycled locally following DNA replication. However, because the experiment was performed using asynchronous cells, it is almost impossible to discern whether the reduction of H3.3-SNAP signals is due to dilution and/or loss of H3.3 through histone exchange. In the revised manuscript, we described the results and discussed the limitation for the interpretation of the results (p17-18).

Reviewer #2

3. *Mcm2* is an DNA helicase and one of key component for replication fork. It is possible that *Mcm2* mutant induced defect in DNA replication first, which consequently resulted in the observed decrease in parent H3.1/H3.3-SNAP amount following DNA replication. Same question is for *Pole3* and *Pole4* mutant as well. How to distinguish between replication and chaperone functions of *Mcm2*, *Pole3* and *Pole4*?

Response: The reviewer's concern about defects in DNA replication is valid. However, I would like to point out the following facts. First, studies from yeast and mammalian cells from ours and others have shown that the *Mcm2-2A* mutant, which contains mutations at two tyrosine residues at histone binding motif of *Mcm2*, does not affect DNA synthesis using a variety of approaches including BrdU-IP-ssSeq, which can measure the relative amount of DNA synthesis at leading and lagging strands [6-8]. Similarly, studies in yeast and mouse ES cells have shown that depletion of *Pole3* and *Pole4* had no detectable defects in DNA synthesis [9-12]. Second, in the revised manuscript, we used FUCCI system to monitor cell cycle progression. We found that the average time for wild type, *Mcm2-2A* and *Pole4* KO cells to transition G1/S to mitosis is quite similar (new Fig. 4b). Third, in our analysis of the distribution of H3.1- and H3.3-SNAP at replicating chromatin using eSPAN, we normalized the eSPAN signals against BrdU-IP-ssSeq signals. This normalization will, in principle, largely eliminate the contribution of defects in DNA synthesis to defects in nucleosome assembly. Together, it is unlikely that defects in parental histone transfer detected in *Mcm2-2A*, *Pole3* KO and *Pole4* KO cells arises from defects in DNA replication. However, we also acknowledge that we cannot rule out this possibility completely.

Reviewer #2

4. SNAP has molecular weight of ~20kD. It may interfere with binding of H3.1/H3.3-SNAP to chromatin proteins and deregulate gene expression. This needs to be taken account when RNA-seq data is analyzed.

Response: The reviewer's point is well taken. To analyze the effects of *Mcm2-2A*, *Pole3* KO on gene expression, we used H3.1-SNAP tagged WT cells as controls. We described this in the experimental procedures.

Reviewer #2

5. Abstract and Introduction: the conclusions were drawn from the experiments using SNAP-tagged system and it needs to be pointed out in the Abstract and Introduction.

Response: Thank you for this suggestion. We have included the SNAP-tagged system description in the abstract and introduction accordingly.

Reviewer #2

6. Abstract: “same mechanisms” needs to be replaced, as while Mcm2, Pole3 and 4 are shared by H3.1 and H3.3, we do not know yet if other chaperones are used by H3.1 and H3.3 differently.

Response: Thank the reviewer for the suggestion. We have now replaced the “same mechanisms” with “shared mechanisms”.

Reviewer #2

7. Line 13 in page 3: “telomeric heterochromatin”. H3.3 is also localized at other heterochromatin regions as well.

Response: Thanks for pointing out. We have now addressed this sentence with “as well as heterochromatin regions”.

Reviewer #2

8. Figs 2, 3 and 5: double mutants caused more severe mitotic defects than single mutant, whereas no difference of H3.1/H3.3 recycling was seen between double and single mutant. Why?

Response: This is a question that is also raised by reviewer #1. As stated above, one possible reason for this result is due to the sensitivity of our assays. To compound the effects, we observed significant more cell death in Mcm2-2A+Pole3 KO and Mcm2-2A+Pole4 KO cells double mutant cells than either single mutant alone (new Supplementary Fig. 3d). We speculate that double mutant cells with a dramatic reduction in parental histone recycling will undergo apoptosis and could not be detected by live cell imaging. We discussed these points in the result section (p10).

Reviewer #2

9. Supplementary Figure 1D: are any chromosome aberrations were observed? How about cell cycle?

Response: We did not see chromosome aberrations in tagged cell lines. We have also analyzed the cell cycle of H3.1-SNAP and H3.3-SNAP cells and compared with E14 WT cells. We did not observe any changes in cell cycle progression of these lines expressing H3.1- and H3.3-SNAP tag compared to wild type cells (new Supplementary Fig. 1f,g).

Reviewer #2

10. Supplementary Figure 5h. the lines and names in the legend are not aligned. Please correct.

Response: Thank you for pointing out this mistake. We have now corrected it.

Reviewer #3 (Remarks to the Author):

In the manuscript titled “Stable inheritance of H3.3-containing nucleosomes during mitotic cell divisions”, Xiaowei Xu et al tracked the dynamic of parental histone H3.1 and H3.3 throughout

one cell cycle using the live cell imaging. Overall their results showed that Mcm2 and Pole3/Pole4 regulate the recycling of histone H3.1 and H3.3 during DNA replication and maintain the genome integrity. The manuscript revealed the transfer of parental H3.1 and H3.3 is mediated by the same mechanism, which contributes to the epigenetic memory and the maintenance of genome stability. However, there are still some ambiguities in this manuscript, and some concerns need to be addressed before this manuscript is considered to publish.

Response: We thank the reviewer for his/her time to review this manuscript and for his/her insightful comments.

Reviewer #3

1. The authors used the fluorescent staining to quantify the recycle of parental H3 during the DNA replication processes. However, how to rule out the fluorescent labelled histones, which are evicted from chromatin, are not assembled or deposited into chromatin during the DNA replication process? The authors may perform immunofluorescent staining, in which the free fluorescent histone can be washed away, to rule out the free fluorescent labelled histones.

Response: We thank the reviewer's suggestions. I agree with the reviewer that analysis of H3.3-SNAP signals using live cell imaging cannot exclude the possibility that once evicted, some parental H3.3 will not be assembled into nucleosomes following DNA replication. However, I would like to point out that the detection of parental H3 using IF after extraction needs cell fixation, which will make it impossible to monitor parental H3.3 recycling in the same cell at different stage of the cell cycle. Therefore, while the IF analysis takes into consideration of the potential free H3.3-SNAP that are not assembled into nucleosomes, the IF results will be represented by the average of H3.3 signals in different cells and likely at different stage of the cell cycle. Second and importantly, while it is possible that parental H3.3-SNAP, once evicted, may not be assembled into nucleosomes, to our knowledge, little, if any, experimental evidence support the hypothesis that a large fraction of parental H3.3 are not re-assembled into nucleosomes. In fact, it has been reported that excess free histone H3, if not assembled into nucleosomes, are subjected to degradation [13].

Reviewer #3

2. The authors performed parental H3.3-SNAP CUT-Tag on nascent DNA, we wonder whether the distribution of parental H3.3 have specificity on the genome compared with total H3.3. Moreover, are the parental H3.3 immediately restored to the half of the level before replication or restored gradually? Are parental H3.3 well preserved at all peak regions during the cell cycle or only at some specific genomic regions? We recommend that the authors had better to perform time-scale (chase for different time after BrdU pulse) parental H3.3-SNAP CUT-Tag to analyze whether there are some specific patterns for the recycling of parental H3.3 during cell cycle, and find out which regions are regulated by Mcm2 or Pole3/Pole4.

Response: The reviewer raised several questions, and let me answer them one by one. To address whether the distribution of parental H3.3 has specificity compared to total H3.3, we performed H3.3-SNAP-Biotin CUT&RUN at different time points (0 hr, 5 hr and 11 hr) following labeling H3.3-SNAP with biotin and compared parental H3.3-SNAP CUT&RUN (1 hr) profiles with H3.3-SNAP CUT&RUN (0h). We observed that the distribution pattern of

H3.3 at 11 hrs is the same as 0 hr based on multiple analysis (Fig. 6). These results are consistent with the idea that H3.3 is likely recycled locally following DNA replication. Moreover, these results also suggest that parental H3.3 are well preserved at all peaks regions.

It would be very interesting to determine whether parental H3.3 is immediately restored or restored gradually following DNA replication. However, it is very challenging to address this question without the usage of sophisticated mass spectrometry analysis. Therefore, I think that it is out of the scope of the present study to address this interesting question.

Finally, I agree with the reviewer that it would be interesting to identify regions of H3.3 affected by Mcm2, Pole3 and Pole4 mutation using H3.3-SNAP-biotin CUT&Tag. However, our initial analysis did not identify H3.3 specific regions during cell cycle.

Reviewer #3

3. This manuscript revealed a portion of H3.3 has to be recycled during DNA replication, what genes are regulated by these “replication-coupled H3.3”?

Response: This is an interesting question. As shown above, the distribution pattern of parental H3.3 is the same as total H3.3 following DNA replication. Therefore, it is very challenging to address this question.

Reviewer #3

4. The conclusion “All together, these results suggest that faithful recycling of parental H3.3 is required for maintenance of genomic integrity during cell division.” in page 13 is not accurate. Data in this manuscript cannot rule out the defects in the Mcm2 and Pole3/Pole4 KO cells is caused by the disturbed recycling of parental H3.1. So we recommend use the word “parental histone” instead of “parental H3.3”.

Response: We agree with the reviewer and modify the text accordingly throughout the text.

Reviewer #3

5. For figure 5 C, we would recommend adding scale bar.

Response: Thank you for pointing this out. We have now added the scale bar at new Fig. 7c (old Fig. 5c).

Reviewer #3

6. “Both Mcm2 and Pole4 coimmunoprecipitated with H3.3-Flag, confirming that that they interact with histone H3.3 in vivo (Fig. 4a)”. Please notes: the “that” is redundant.

Response: Thank the reviewer for pointing this out. We have now edited it accordingly.

References:

1. Zasadzinska, E., et al., *Inheritance of CENP-A Nucleosomes during DNA Replication Requires HJURP*. *Dev Cell*, 2018. **47**(3): p. 348-362 e7.
2. Nechemia-Arbely, Y., et al., *DNA replication acts as an error correction mechanism to maintain centromere identity by restricting CENP-A to centromeres*. *Nature Cell Biology*, 2019. **21**(6): p. 743-754.
3. Mendiratta, S., A. Gatto, and G. Almouzni, *Histone supply: Multitiered regulation ensures chromatin dynamics throughout the cell cycle*. *J Cell Biol*, 2019. **218**(1): p. 39-54.
4. Garibaldi, A., F. Carranza, and K.J. Hertel, *Isolation of Newly Transcribed RNA Using the Metabolic Label 4-Thiouridine*. *Mrna Processing: Methods and Protocols*, 2017. **1648**: p. 169-176.
5. Xu, M., et al., *Partitioning of Histone H3-H4 Tetramers During DNA Replication-Dependent Chromatin Assembly*. *Science*, 2010. **328**(5974): p. 94-98.
6. Gan, H., et al., *The Mcm2-Ctf4-Polalpha Axis Facilitates Parental Histone H3-H4 Transfer to Lagging Strands*. *Mol Cell*, 2018. **72**(1): p. 140-151 e3.
7. Petryk, N., et al., *MCM2 promotes symmetric inheritance of modified histones during DNA replication*. *Science*, 2018. **361**(6409): p. 1389-1392.
8. Li, Z., et al., *DNA polymerase alpha interacts with H3-H4 and facilitates the transfer of parental histones to lagging strands*. *Sci Adv*, 2020. **6**(35): p. eabb5820.
9. Araki, H., et al., *Cloning DPB3, the gene encoding the third subunit of DNA polymerase II of Saccharomyces cerevisiae*. *Nucleic Acids Res*, 1991. **19**(18): p. 4867-72.
10. Ohya, T., et al., *Structure and function of the fourth subunit (Dpb4p) of DNA polymerase epsilon in Saccharomyces cerevisiae*. *Nucleic Acids Res*, 2000. **28**(20): p. 3846-52.
11. Siamishi, I., et al., *Lymphocyte-Specific Function of the DNA Polymerase Epsilon Subunit Pole3 Revealed by Neomorphic Alleles*. *Cell Rep*, 2020. **31**(11): p. 107756.
12. Yu, C., et al., *A mechanism for preventing asymmetric histone segregation onto replicating DNA strands*. *Science*, 2018. **361**(6409): p. 1386-1389.
13. Wike, C.L., et al., *Excess free histone H3 localizes to centrosomes for proteasome-mediated degradation during mitosis in metazoans*. *Cell Cycle*, 2016. **15**(16): p. 2216-2225.

REVIEWERS' COMMENTS

Reviewer #1 (Remarks to the Author):

The authors have significantly addressed my concerns and as such i recommend publication

Reviewer #2 (Remarks to the Author):

The authors have addressed most of my concerns in their revised version of the manuscript.

Reviewer #3 (Remarks to the Author):

In the revised manuscript, the authors address all the questions or concerns that I raised, thus I recommend to accept the manuscript.